# Imaging Evaluation of Ovarian Masses in a Pediatric Population: A Comprehensive Overview

**DOI:** 10.3390/cancers17142316

**Published:** 2025-07-11

**Authors:** Charis Bourgioti, Marianna Konidari, Anastasia Giantsouli, Afroditi Kafritsa, Vassilis Xydis, Lia-Angela Moulopoulos, Maria I. Argyropoulou, Athina C. Tsili

**Affiliations:** 11st Department of Radiology, School of Medicine, National and Kapodistrian University of Athens, Aretaieon Hospital, 11528 Athens, Greece; chbourg@med.uoa.gr (C.B.); mkonidari@med.uoa.gr (M.K.); lmoulop@med.uoa.gr (L.-A.M.); 2Department of Clinical Radiology, Faculty of Medicine, School of Health Sciences, University of Ioannina, University Campus, 45110 Ioannina, Greece; ngiantsouli@gmail.com (A.G.); afroditikafritsa@gmail.com (A.K.); vgxydis@uoi.gr (V.X.); margyrop@uoi.gr (M.I.A.)

**Keywords:** children, diagnostic imaging, magnetic resonance imaging, pediatrics, ovarian neoplasms, ultrasonography

## Abstract

The majority of abdominal masses in the pediatric population originates from the ovaries and includes both non-neoplastic lesions and neoplastic tumors. The incidence, histologic patterns, and clinical presentation of ovarian masses in the pediatric age group differ from those seen in adults. Most ovarian tumors in the pediatric cohort are benign, but malignancies are observed in 3–8% of children and adolescents with an adnexal mass. Imaging findings in combination with clinical characteristics and laboratory results greatly help the characterization of ovarian masses in the pediatric population. Transabdominal US represents the imaging modality of choice for the initial assessment of ovarian masses in a pediatric age. MRI or CT may be used as supplemental tools, in cases of inconclusive US findings. In this narrative review, we aim to present the imaging findings of ovarian masses in the pediatric population, including key points for differential diagnosis.

## 1. Introduction

Ovarian tumors are rare in the pediatric population, including children (ages 0–14 years) and adolescents (ages 15–19 years), with an annual incidence of 2.6 cases per 100,000 girls [1,2,3,4,5,6,7,8,9]. However, they represent the more common neoplasms of the female genital tract in the pediatric cohort, accounting for approximately 60–70% of gynecologic malignancies in this age [10,11,12]. Most pediatric ovarian masses are benign; however, malignancies are reported in 3–8% of cases, primarily in adolescent girls, and account for 1–3% of all childhood cancers [1,2,3,4,5,6,7,8,9,10,11,12]. The incidence, histologic distribution, and clinical presentation of ovarian masses in the pediatric population differ significantly from those in adults [1,2,3,4,5,6,7,8,9].

Ovarian masses in the pediatric age comprise a heterogeneous group, including non-neoplastic and neoplastic lesions. Common non-neoplastic ovarian lesions in children and adolescents include functional cysts, ovarian torsion, and pelvic inflammatory disease (PID). Neoplastic tumors include germ cell tumors (GCTs), sex cord–stromal tumors (SCSTs), epithelial tumors, and miscellaneous tumors, which account for approximately 60–80%, 10–20%, 15–20%, and less than 5% of cases, respectively [1,2,3,4,5,6,7,8,9,10]. Epithelial tumors are the most common type of ovarian neoplasms in adults, but in the pediatric cohort, GCTs are more prevalent. Among these, mature teratomas are the most frequently occurring ovarian neoplasms in this age group.

The combination of clinical manifestations, laboratory data, and imaging findings greatly helps in narrowing the differential diagnosis of ovarian masses in children and adolescents [1,2,3,4,5,6,7,8,9]. Abdominal pain and/or a palpable mass are the primary clinical signs. An acute onset of pain or tenderness may be the symptom in cases of complications, such as torsion, hemorrhage, or rupture. Hormone-secreting ovarian tumors can present with endocrine manifestations including precocious puberty, menstrual irregularities, or signs of virilization [1,3,4,5]. When an ovarian tumor is detected in children, it is important to consider the possibility of a cancer predisposition syndrome, and genetic testing with a thorough family history assessment should follow [6].

Measurements of serum tumor markers can aid in diagnosing cancer, monitoring treatment response, and detecting tumor recurrence. The most clinically valuable tumor markers include alpha-fetoprotein (AFP), beta-human chorionic gonadotropin (β-hCG), lactate dehydrogenase (LDH), cancer antigen 125 (CA 125), and inhibin. Elevated serum levels of AFP, β-hCG, or CA 125 are strongly indicative of ovarian cancer; however, negative results do not rule out the presence of malignancy, as tumor markers are elevated in only 50% of malignant tumors [1,3,4].

Pediatric ovarian sex cord–stromal tumors and epithelial tumors are staged based on the International Federation of Gynecology and Obstetrics (FIGO) staging system [2,13]. Germ cell tumors are staged according to the Children’s Oncology Group (COG) system [2]. Treatment of pediatric ovarian masses needs to be minimally invasive considering, when feasible, fertility preservation. Referral to specialized centers and a multi-disciplinary approach are essential to prevent unnecessary oophorectomies and guarantee the best possible outcome [1,3,14,15,16].

The initial task of imaging in the pediatric age group with a suspected ovarian mass is to determine the precise origin and differentiate between ovarian and extraovarian lesions. Ovarian masses should then be characterized as non-neoplastic or neoplastic. In cases of neoplastic tumors, assessing the likelihood of malignancy is crucial for determining the appropriate treatment approach, either through conservative management or surgery aimed at preserving fertility [4].

Transabdominal US is the modality of choice for the initial imaging of ovarian masses in the pediatric population, due to its widespread availability, ease of use, low cost, and lack of ionizing radiation and sedation risks. In sexually active adolescents, transvaginal US may also be used [1,2,4,6,8,17,18,19,20]. Sonographic findings that favor malignancy include the large size and presence of intratumoral solid components [1,2,6,17,18,19,20]. However, although ovarian tumor size may play a role in the initial risk of a malignancy assessment in the pediatric population, it should be emphasized that at least some small masses, especially those presenting as solid-cystic and/or solid tumors on imaging and regardless of the serum tumor marker expression, can turn out to be malignant.

Although imaging-based scoring systems, such as the Risk of Malignancy Index (RMI), Risk of Ovarian Malignancy Algorithm (ROMA), International Ovarian Tumor Analysis (IOTA) models, and the Ovarian–Adnexal Reporting and Data System (O-RADS), are widely used to characterize and stratify the risk of malignancy in ovarian masses in the adult population, their application in the pediatric population is significantly less documented [21,22,23,24,25,26,27,28]. These diagnostic systems integrate imaging findings (primarily sonographic), tumor markers, and clinical factors (e.g., menopausal status) to differentiate between benign and malignant adnexal masses. However, their applicability and validation in the pediatric age group remain limited. This is primarily due to the rarity of ovarian masses in children and adolescents, as well as the distinct histopathologic profile in this age group. In pediatric patients, benign lesions—such as functional cysts and mature teratomas—are far more common, while malignancies are rare. When malignancies do occur, GCTs like dysgerminomas are the most frequent types, in contrast to the adult population, where epithelial ovarian cancers predominate.

Specifically, the RMI and the ROMA scoring systems have not been validated in the pediatric age cohort. Moreover, their dependence on menopausal status and serum CA 125 levels limits their utility in this age group.

The IOTA group has developed several diagnostic models—including Simple Rules, logistic regression models (LR1 and LR2), and the Assessment of Different Neoplasias in the Adnexa (ADNEX) model—to classify adnexal masses as benign, malignant, or indeterminate. These models incorporate standardized sonographic features, such as the presence of solid components, Doppler flow characteristics, and ascites. In the adult population, IOTA models have demonstrated superior diagnostic performance compared to traditional tools such as the RMI and the ROMA. Despite their widespread validation in adults, data on the applicability of IOTA models in the pediatric population remain limited. Preliminary studies indicate that the use of standardized US descriptors enhances diagnostic accuracy in younger patients. Notably, IOTA Simple Rules and Benign Descriptors have shown promising sensitivity and specificity in this demographic [12,25,27]. Nevertheless, further adaptation and robust validation are essential before these models can be reliably implemented in pediatric clinical practice.

The O-RADS US system, developed by the American College of Radiology (ACR), offers a standardized lexicon and risk stratification framework for the evaluation of adnexal masses using US. The system categorizes lesions from O-RADS 1 (normal) to O-RADS 5 (high risk of malignancy), facilitating consistent interpretation and clinical decision-making [23,24]. A recent study investigating the application of O-RADS US in the pediatric population has demonstrated high diagnostic accuracy, substantial inter-observer agreement, and effective risk stratification [28]. These findings support the potential utility of the O-RADS as a reliable and objective tool for the preoperative assessment of ovarian masses in children and adolescents [2,28]. Nonetheless, additional prospective, multicenter studies are warranted to further validate its performance and generalizability in this demographic.

Contrast-enhanced US (CEUS) is increasingly used for the characterization of ovarian masses by assessing their vascularity and enhancement patterns. The combination of the O-RADS with CEUS has been shown to improve the diagnostic accuracy of sonography in the characterization of adnexal masses in the adult population [29,30]. Preliminary observations suggest that conventional US combined with CEUS significantly aids in differentiating benign from malignant pediatric ovarian masses, while enhancing the diagnostic performance of junior radiologists [31,32].

MRI of the abdomen and pelvis represents a valuable complimentary imaging technique for lesion characterization, staging, preoperative planning, and/or follow-up [2,4,8,33,34,35,36,37,38,39,40,41]. MRI is the preferred imaging modality for the assessment of sonographically indeterminate ovarian masses, due to its excellent soft-tissue contrast, its detailed anatomic and functional information, and the absence of radiation exposure [4,34,35,36]. The sensitivity and specificity of the technique in the characterization of pediatric ovarian masses vary between 84.8 and 100%, and 20 and 98.4%, respectively [33]. Based on the results of a recently published retrospective study, the O-RADS MRI system can be effectively used in pediatric patients [36]. CT is an acceptable alternative modality in children who cannot undergo MRI [2,4]. CT is also recommended in emergency situations [8].

This narrative review aims to provide a comprehensive overview of the imaging findings of ovarian masses in the pediatric population.

## 2. Search Strategy and Study Selection

A comprehensive literature search was performed for all publications that reported the imaging findings of ovarian masses in the pediatric population from the PubMed/MEDLINE database and included articles published from 1977 to April 2025.

The following keywords were used: “imaging”, “ovarian tumors”, ovarian masses”, and “pediatric”. Citations and references of the retrieved studies were used as additional sources. Only papers in English language were assessed.

The flow chart of the selection process is shown in Figure 1 [1,2,3,4,5,6,8,9,12,13,14,16,17,18,19,20,23,27,28,31,32,33,36,37,38,39,40,41,42,43,44,45,46,47,48,49,50,51,52,53,54,55,56,57,58,59,60,61,62,63,64,65,66,67,68,69,70,71,72,73,74,75,76,77,78,79,80,81,82,83,84,85,86,87,88,89,90,91,92,93,94,95,96,97,98,99,100,101,102,103,104,105,106,107,108,109,110,111,112,113,114,115,116,117,118,119,120,121,122,123,124,125].

## 3. Non-Neoplastic Ovarian Lesions

### 3.1. Functional Cysts

Functional or physiological cysts are prevalent across all age populations, including the pediatric age group, due to various hormonal stimulations. Functional cysts include follicular and corpus luteum cysts and occur during the normal menstrual cycle. In the pediatric population, they account for approximately 60% of all ovarian masses [3,112,126]. Causative factors include the influence of maternal and placental hormones during infancy, the secretion of gonadotropins by the developing pituitary gland before puberty, and irregular ovulation during adolescence. These lesions are typically asymptomatic and tend to resolve spontaneously [3,112,126].

In young children, normal ovaries have an ellipsoid shape, appear solid and homogeneous, and may contain primordial follicles measuring up to 9 mm in diameter that usually regress spontaneously (Figure 2 and Figure 3) [61,72,86,88,126]. In pubescent girls, ovarian follicles measure up to 2.5 cm; cysts larger than 4 cm generally require imaging follow-up at two or six weeks to confirm resolution [61,72,86,88].

Follicular cysts develop after a failure of release of the ovum, either due to excessive stimulation by the follicle-stimulating hormone or the absence of the midcycle surge of the luteinizing hormone, which promotes a rupture of the dominant follicle during the process of ovulation. These cysts are usually unilocular, smooth, thin-walled, and typically larger than 3 cm, but not exceeding 10 cm in diameter.

On US, follicular cysts are predominantly simple, unilocular cysts with thin walls, anechoic content, posterior acoustic enhancement, and an absence of internal or peripheral vascularity (Figure 4) [3,61,126,127]. MRI is generally not necessary; however, it is recommended for further evaluation of any cystic lesion larger than 7.5 cm, not visualized in its entirety, to avoid missing the presence of solid tissue [2].

Corpus luteum cysts form after a failure of regression of the corpus luteum. During the normal menstrual cycle and in the absence of pregnancy, the functional lifespan of the corpus luteum is approximately 14 days, after which it involutes to form the corpus albicans. Corpus luteum cysts present as either simple or complex cystic masses, usually less than 3 cm in diameter, typically with thick, crenulated walls and potential internal debris [3,126].

Sonographically, they are characterized by thick walls and hyperechoic content. They typically demonstrate increased peripheral blood flow on Doppler US, a sign called the “ring of fire”. Due to small amounts of internal hemorrhage, they may display a characteristic spider-web-like appearance; they can also contain blood clots that may mimic solid components. However, these clots, unlike solid tissue, lack blood flow and typically display concave borders and a characteristic jelly-like “wobbling” motion within the cyst when the US transducer is utilized to gently manipulate the ovary [3,61,126,127]. On MRI, corpus luteum cysts demonstrate variable signal intensity and ragged internal walls, with a characteristic peripheral ring of restricted diffusivity and intense contrast enhancement (Figure 3, Figure 5 and Figure 6) [128].

### 3.2. Hemorrhagic Cysts

Hemorrhagic cysts frequently develop as a complication of functional cysts and they typically regress on follow-up imaging, after one or two menstrual cycles. They represent a frequent cause of acute pelvic pain [112]. Depending on the age of blood products, they present with variable US appearances. In the acute phase, ovarian enlargement may be the only sonographic abnormality, due to the similar echogenicity of the hemorrhagic elements and the surrounding ovarian stroma. With time, hemorrhagic cysts characteristically appear as complex cystic masses, with thin, linear internal echoes, corresponding to fibrin and fluid–fluid levels. These linear strands create a reticular or “fishnet” appearance. Doppler US does not reveal internal blood flow (Figure 2 and Figure 7) [126,127]. In indeterminate cases, MRI can accurately differentiate hemorrhagic cysts from neoplastic masses. On MRI, hemorrhagic cysts are typically hyperintense on T1-weighted imaging (T1WI), without a signal drop on fat suppression sequences, and heterogeneously hypointense on T2-weighted imaging (T2WI), with an absence of contrast enhancement (Figure 2 and Figure 8) [128,129,130].

### 3.3. Ovarian Torsion

Ovarian torsion occurs when either the ovary, the fallopian tube, or both rotate around their ligaments. Most of the time, both the ovary and a part of the ipsilateral fallopian tube are involved; therefore, the term adnexal torsion may be more accurate. Torsion of the vascular pedicle causes obstruction of the lymphatic, venous, and finally arterial flow, ultimately leading to infarction and necrosis. Ovarian torsion in the pediatric population accounts for 15% of all cases. Although it can occur in all pediatric age groups, about half of the cases (52%) occur in the perimenarchal period, followed by 16% of cases in infants younger than one year [3,59,108,112,113,114,115,126].

In the majority of cases (51–84%) with ovarian torsion in the pediatric cohort, there is an underlying adnexal pathology that serves as a lead point, usually of benign origin and, most frequently, a follicular cyst or a mature teratoma [3,43,108,112,113,114,126]. However, torsion can occur in normal adnexa more frequently in pediatric patients compared to in adults. This may be due to the greater length and laxity of ligaments in some young females which allows for increased adnexal mobility [3,43,108,112,113,115,126]. Ovarian torsion is rarely associated with pelvic inflammatory disease and endometriosis, probably due to the development of adhesions. Malignant neoplasms are rarely a cause of torsion, which may be explained by their large size and the greater gravitational force of their solid components that restrict ovarian mobility in the shallow pubertal pelvis [106].

Imaging has a critical role in the diagnosis of ovarian torsion in pediatric patients, as the history and clinical presentation are often nonspecific. Ultrasound is the modality of choice when there is clinical suspicion of ovarian torsion [3,108,112,113,115,116,117,126,127,131,132]. The most common US finding is a unilateral, asymmetric increase in ovarian volume (Figure 9 and Figure 10). A useful tip is to compare the size of both ovaries in order to evaluate whether the ovary at the affected side is enlarged. If the ovarian volume is at least three times greater than the expected size in relation to age, torsion should be suspected [108]. The enlarged ovary has variable US appearances, related to the degree of internal hemorrhage, stromal edema, infarction, and necrosis, and may appear solid in early torsion and heterogeneous or cystic in later phases. A specific sonographic sign seen in cases of ovarian torsion in adolescents is the presence of multiple follicles at the periphery of a unilaterally enlarged edematous ovary, a finding resembling “a string of pearls”. The cysts are often spherical and smooth-walled, with a diameter ranging from 8 to 15 mm to 25 mm and may present fluid–debris levels (Figure 9). Ovarian torsion should be strongly suspected in a girl presenting with unilateral pelvic pain, when imaging reveals an asymmetrically enlarged ovary with peripherally located cysts. Other imaging findings include displacement of the twisted ovary towards the midline or near the uterine fundus, deviation of the uterus toward the affected ovary, and free pelvic fluid [3,108,112,113,115,116,117,126,127,131,132].

A twisted pedicle is a hallmark feature of ovarian torsion, resulting from edema and congestion in the twisted blood vessels, fallopian tubes, and supporting ligaments. It appears as an amorphous or tubular structure, often with a beak-like shape, situated between the uterus and the twisted ovary or adnexal mass. The combination of an adnexal mass; a twisted pedicle, referred to as the whirlpool sign; and the absence of blood flow upon Doppler examination is highly suggestive of torsion [3,108,112,113,116,126]. However, detection of an arterial flow on Doppler US cannot safely exclude adnexal torsion. This is because of the dual adnexal supply from both the ovarian and uterine arteries and because arterial obstruction occurs after a lymphatic and venous drainage compromise [3,112,113,126]. Moreover, in cases of incomplete torsion, the blood flow may be still evident upon Doppler examination. On the other hand, in the pediatric population, normal ovaries may demonstrate a low or absent flow; thus, this finding should be interpreted with caution and in the appropriate clinical setting [3,112,113,126].

In pediatric patients with suspected torsion and inconclusive US examination, second-line imaging with CT or preferably MRI should be considered if it does not delay potential surgery [3,59,108,112,113,126,132,133,134,135]. Ovarian torsion is a surgical emergency and prompt intervention is crucial; therefore, time should not be wasted in establishing an imaging diagnosis if the clinical suspicion is high. The advantage of CT is that it can be performed on an emergency basis. However, unlike US and MRI, it exposes children and adolescents to ionizing radiation.

CT and MRI findings of ovarian torsion are similar to US (Figure 9 and Figure 10). Contrast-enhanced imaging should be obtained to evaluate ovarian stromal enhancement, which ranges from reduced to absent and may be associated with a hemorrhage and necrosis (Figure 9 and Figure 10) [118,133,134,135]. Post-contrast imaging may also aid in detecting the swirling appearance of a twisted pedicle [134]. Regarding diffusion-weighted imaging (DWI), studies have shown that a lower apparent diffusion coefficient (ADC) and higher DWI signal of either the ovarian stroma or the thickened fallopian tube compared to normal restriction of the ovarian stroma might be indicative of hemorrhagic infarction [118,135,136,137].

### 3.4. Pelvic Inflammatory Disease/Tubo-Ovarian Abscess

Pelvic inflammatory disease is the result of an infection of internal genitalia, most commonly by Chlamydia trachomatis and Neisseria gonorrhoeae, and mainly affects sexually active females, usually between the ages of 15 and 25 years [3]. A tubo-ovarian abscess (TOA) and pyosalpinx are adnexal manifestations of the disease. A tubo-ovarian abscess most commonly appears as a complex, thick-walled, fluid-filled lesion with internal septa. Pyosalpinx manifests as a fluid-filled tubular adnexal structure with thickened walls [3].

On US, TOAs may demonstrate mixed echogenicity or have a ground-glass appearance [3]. On MRI, the content of the mass exhibits variable signal intensity depending on the amount of proteinaceous and hemorrhagic products. It is usually heterogeneously hyperintense on T2WI and of variable, usually low, signal intensity on T1WI [138]. The presence of gas is pathognomonic of a TOA; however, it is found only in 22–38% of cases [139]. The walls and septa typically show a low signal on both T1WI and T2WI and prominent enhancement [3]. On T1WI, an internal hyperintense halo with avid enhancement may be observed, due to the combination of a hemorrhage and granulation tissue. T2 shading at the periphery of the abscess may also be observed [140]. On DWI, acute TOAs demonstrate high signal intensity on high-*b*-value images and low signal intensity on the corresponding ADC map, indicative of restricted diffusion of the purulent content [141]. Typically, in continuity with a TOA, a pyosalpinx is present, seen as a tubular, cystic structure with thick, enhancing walls (Figure 11). Additional findings include adnexal edema; inflammatory changes in the surrounding structures; and, in chronic cases, adhesions, which appear as mesh-like, linear strands of a low T1 signal, with mild enhancement [140].

## 4. Neoplastic Ovarian Lesions

### 4.1. Germ Cell Tumors

Germ cell tumors are the most common histologic type in the pediatric age group, accounting for 60–80% of all ovarian neoplasms in this population, whereas epithelial tumors are more prevalent in adults [1,3,6,49,125,142]. According to the World Health Organization classification, histologic types include a teratoma (mature and immature), dysgerminoma, yolk sac tumors, embryonal carcinoma, polyembryoma, non-gestational choriocarcinoma, and mixed GCTs [1,3,142]. The majority of GCTs are benign, with a mature teratoma being the most commonly encountered subtype. One-third of ovarian GCTs in children and adolescents are malignant, with dysgerminoma being the most frequent malignant ovarian germ cell tumor (MOGCT) [1,3,6,125,142].

The prognosis for ovarian GCTs in the pediatric population is generally favorable, even for MOGCTs. The main prognostic factors include the stage of the disease at diagnosis, histologic type, and grade and serum tumor marker levels, such as AFP and β-HCG [1].

#### 4.1.1. Mature Teratoma

A mature teratoma, also known as a mature cystic teratoma (MCT) or a dermoid cyst, is the most common GCT and the most common benign ovarian tumor in the pediatric age group, especially in the first two decades of life, accounting for more than half of all cases in this population [1,3,4,16,71,121,122,142]. It is mostly asymptomatic and typically an incidental finding; however, complications include torsion (3–16%), rupture (1–4%), and less likely infection (1%) [1,3,4,142]. Malignant transformation, often to squamous cell carcinoma, is rare in the pediatric population and mostly occurs in postmenopausal women with large, long-standing tumors [1,3,4,142]. In 10–25% of cases, bilateral MCTs are present [1]. A mature teratoma may also coexist in the contralateral ovary in approximately 10% of patients with MOGCTs [1]. The primary treatment of an MCT in the pediatric population is surgical excision, with cystectomy and laparoscopic surgery [1,3,4].

Mature teratomas are composed entirely of mature tissues, originating from two or three germ layers, including the ectoderm, mesoderm, and endoderm. Mature cystic teratomas are typically well-circumscribed, encapsulated cystic lesions; are most commonly unilocular; and contain macroscopic fat. A hallmark of these tumors is the presence of a Rokitansky nodule or dermoid plug—a solid protuberance projecting into the cystic cavity. This nodule often harbors a variety of tissue components, including hair, bone, teeth, muscle, cartilage, and calcifications [1,3,4,142].

Ultrasound is the first-line modality in the investigation of mature teratomas [1,3,4,56,121,143,144,145,146]. Sonographic appearance varies depending on the internal composition of the lesion. Typically, it presents as a unilocular cystic mass with an absence of internal vascularity and the presence of a hyperechoic Rokitansky nodule that arises from the cyst wall and protrudes into the lumen. This nodule frequently contains hair follicles, which may appear as echogenic dots or strands on US. Additionally, it often has calcified elements such as teeth or bone, resulting in posterior acoustic shadowing. When the nodule is located superficially or extends to the lesion’s surface, it can obscure the visualization of the lesion’s deeper components and posterior wall due to this shadowing. This sonographic characteristic is commonly known as the “tip of the iceberg” sign [147]. Additional US findings include the following: fat–fluid levels, likely due to the layering of fatty sebum atop the serous fluid; the “dermoid mesh” sign, which reflects the presence of multiple hyperechoic lines/bands and dots corresponding to floating hair within the cyst; and an echogenic mass, indicative of sebaceous material mixed with echogenic floating debris (Figure 12 and Figure 13) [1,3,4,121,143,144,145,146].

The sonographic appearance of the mature teratoma may be misinterpreted as bowel loops containing gas or fecal material, hemorrhagic cysts, or endometriomas. CT or MRI should be considered in equivocal cases, because of their better spatial resolution and greater sensitivity in fat detection [1,3,4,56,91,148,149]. Large amounts of fat or coarse calcifications within the tumor are almost pathognomonic (Figure 13); however, in a small minority of mature teratomas, fat may be absent or present only in the cyst wall or in the Rokitansky nodule [1,3,4,91,145].

CT can reveal the presence of fatty components and fat–fluid levels, calcifications, hair, and the Rokitansky nodule within the ovarian tumor, helping to establish an accurate diagnosis in approximately 98% of cases (Figure 13) [1,3,4,91,145,149]. The identification of multiple mobile nodules consisting of varying proportions of sebaceous substance, adipose tissue, keratin, hair, fibrin, and hemosiderin within an ovarian cystic lesion is termed the “floating balls” sign and is considered pathognomonic for a mature teratoma; however, it is identified in merely 3.3% of cases [150]. The presence of a single large mass at the fat–fluid interface is referred to as the “Poké Ball” sign [148].

On MRI, fatty components demonstrate high T1 and T2 signals, with suppression at fat saturation sequences (Figure 12, Figure 14 and Figure 15). T1-weighted imaging with fat suppression is most useful for discriminating fat from a hemorrhage (which remains bright). Fat–fluid levels can also be observed, and a chemical shift artifact is frequently (62–87%) present [1,3,4,145]. In almost a third of cases, macrospopic fat may not be obvious in lipid-poor teratomas. In such cases, opposed-phase T1WI may reveal areas with a signal drop indicative of intracellular fat, and fat-only T1 Dixon sequences may aid in the identification of small foci of fat in the cyst wall or the Rokitanski nodule (Figure 15) [3,4,145]. DWI may also prove helpful, since keratin, which is also present in lipid-poor teratomas, shows restricted diffusion [3,4,143,151]. Calcification or teeth within the mass present with marked hypointensity on all sequences [3,4,145].

The solid components in the Rokitansky nodule may be enhanced after contrast material administration, with a time–intensity curve that is type 1, 2, or even 3 on dynamic-contrast-enhanced (DCE) MRI, depending on the nature of the solid tissue (e.g., thyroidal), without necessarily indicating malignancy [152]. In most cases, a Rokitansky nodule shows peripheral enhancement; however, the presence of large irregular enhancing components and accompanying features, such as invasion of the wall of the cyst, should raise suspicion of malignancy [23]. MRI is useful, but it cannot always safely differentiate between mature and immature teratomas. In some cases, a teratoma may coexist with another GCT subtype, which may not always be identifiable on imaging. In such cases, correlation with tumor markers and complete surgical staging is needed [122].

#### 4.1.2. Immature Teratoma

An immature teratoma is the second most common MOGCT in the pediatric population, accounting for 10–20% of malignant neoplasms in this age population [1,3,4,6]. It is usually diagnosed in a younger age group, with the mean age at presentation being 10 years, and, in fact, the younger the patient, the higher the probability that a teratoma is of the immature type. Ipsilateral or contralateral MCTs are observed in 26% and 10% of immature teratoma cases, respectively [1,3,4,122].

An immature teratoma is a type of GCT, that, like an MCT, arises from more than one germ cell layer. However, it contains both immature and varying amounts of mature tissues. The quantity of immature elements—particularly immature neuroepithelial tissue—determines the histologic grade of the tumor [1,3,4,6,125]. Many cases also contain yolk sac tumor elements and are therefore associated with increased AFP levels in approximately 33–65% of patients [122].

This neoplasm exhibits more aggressive clinical behavior compared to that of a mature teratoma. Despite this, most patients present with early-stage disease and the prognosis is often favorable. Surgery, with unilateral salpingo-oophorectomy and a staging procedure, is the treatment of choice [1,3,4,6].

Immature teratomas radiologically present as unilateral, heterogenous masses, which are larger and more complex when compared to mature teratomas. The mean diameter of immature teratomas is 16 cm, while that of mature teratomas is 6.5 cm; thus, the size of the tumor may aid in differential diagnosis [1,3,4,145]. The US appearance of immature teratomas is nonspecific. These tumors often present as heterogeneous, partially solid, highly vascular masses, with scattered calcifications. Small foci of fat within the solid part may be difficult to detect sonographically [145,153,154]. On MRI, the tumor usually contains extensive solid components, with avid enhancement and only scattered foci of fat and small cystic areas with simple serous fluid, as opposed to the sebaceous material, which is characteristic of an MCT (Figure 16) [145,155]. Detection of a solid part interspersed with multiple cystic areas is a key feature that aids in differentiating immature teratomas from MCTs. Calcifications may be present and are usually multiple, small, and irregular, having a pattern that differs from the coarse or toothlike calcifications of a mature teratoma [1,3,4,14].

Rarely, an immature teratoma may be associated with an uncommon phenomenon, known as gliomatosis peritonei, characterized by implantation of mature glial cells at the peritoneal surfaces or lymph nodes [1,156]. MRI typically shows T2 hyperintense foci within the peritoneal cavity, ascites, and nodules with enhancement [157].

#### 4.1.3. Dysgerminoma

Dysgerminoma accounts for the majority of ovarian malignancies in the pediatric population and constitutes approximately 30% of MOGCTs. It typically affects young females in the second and third decades of life, although approximately 10% of cases occur in the first decade [1,3,4,6,158]. Unlike other MOGCTs, dysgerminomas are bilateral in 10–15% of cases.

Dysgerminoma is considered the ovarian counterpart of testicular seminoma, and it is composed of large, uniform cells with clear cytoplasm and central nuclei, showing no specific differentiation [1,3,4,6,158].

Serum tumor markers, including LDH and β-hCG, may be elevated. LDH is elevated in up to 95% of cases, making it a valuable diagnostic and follow-up marker. Beta-hCG is elevated in approximately 5% of cases, typically in tumors containing syncytiotrophoblastic giant cells [1,3,4,6,158]. Dysgerminoma is the most common ovarian malignancy associated with gonadal dysgenesis and is also associated with abnormal gonads (gonadoblastoma) and chromosomal abnormalities, such as Turner syndrome [1,3,4,6,159].

The prognosis is excellent, especially since most patients are diagnosed at an early stage, making surgical treatment often curative. However, relapse occurs in 13–20% of cases, usually within 19–24 months [1,3,4,6].

Dysgerminoma typically appears as a large, well-circumscribed, lobulated, predominantly solid mass with fibrovascular septa. Areas of necrosis, hemorrhage, cystic change, and calcifications may be present within the neoplasm [1,3,4,6]. Pelvic or retroperitoneal lymph node involvement may be present.

The key diagnostic feature is the presence of fibrovascular septa. On US, dysgerminoma presents as a solid mass with possible areas of necrosis, hemorrhages, and speckled calcifications. The fibrovascular septa are hypoechoic due to their fibrous nature and demonstrate prominent blood flow upon Doppler examination [1,3,4,6,158]. On MRI, the fibrovascular septa usually show a low T2 signal, although not as low as that seen in more fibrous tumors, like stromal or Brenner tumors. After contrast administration, the septa typically exhibit avid, homogeneous enhancement. Areas of necrosis, hemorrhages, cystic change, and calcifications may also be visualized [1,3,4,6,94,160].

#### 4.1.4. Yolk Sac Tumor

Yolk sac tumors, formerly known as endodermal sinus tumors, are the second most common MOGCT in the pediatric population, which usually appears during the second and third decades of life [1,3,4,6]. Yolk sac tumors usually demonstrate aggressive behavior and rapid growth and are often accompanied by ascites and metastatic spread, more often lymphatic, although peritoneal and hematogenous spread may also occur [1,3,4,158]. Yolk sac tumors, along with mixed GCTs, are associated with high levels of AFP [1,3,4,158].

Most cases present as unilateral, early-stage disease and can be managed with conservative surgery and multi-agent chemotherapy [1,3,4].

On pathology, yolk sac tumors typically appear as a large, well-delineated, solid, and cystic unilateral mass with a friable, hemorrhagic, and necrotic appearance [1,3,4].

Imaging appearances of yolk sac tumors may be nonspecific, including unilateral, large, mixed solid, and cystic masses with an intratumoral hemorrhage, necrosis, and prominent contrast enhancement (Figure 17) [1,3,4]. On US, fine-textured slightly hyperechoic solid intratumoral components with rich vascularity are detected [161]. A typical finding on MRI is that of dilated vessels or vascular aneurysms, identified as signal voids on spin-echo sequences or enhancing vessels on contrast-enhanced imaging, the so-called “bright dot” sign [1,3,4,162,163]. Linear tearing of the tumor capsule and a consequent rupture may occur in up to a third of cases [164]. Although imaging findings may be inconclusive, in young patients presenting with a large, predominantly solid ovarian mass and elevated serum AFP levels, the possibility of a yolk sac tumor should be strongly considered [1].

#### 4.1.5. Other Germ Cell Tumors

##### Non-Gestational Choriocarcinoma

Non-gestational choriocarcinoma is an extremely rare MOGCT, occurring in the pediatric age cohort, usually as part of a mixed GCT [1,3,158]. Beta-hCG levels are usually elevated and may be associated with vaginal bleeding and precocious puberty. Surgery combined with chemotherapy is the primary treatment. Prognosis is generally poor, since metastases occur early in the course of the disease [1,3,158]. Imaging characteristics are poorly described [1,3]. On US, the tumor may appear as a complex, well-defined, highly vascular adnexal mass [1,165,166,167].

##### Embryonal Carcinoma

Embryonal carcinoma is a rare, primitive MOGCT of childhood and adolescence, most often identified as a component of a mixed GCT. Pure embryonal carcinoma is extremely uncommon. It is a highly malignant neoplasm, with a mean age at diagnosis of 14 years [1,3,4]. This tumor may secrete AFP and/or β-hCG in up to 60% of cases, which may induce endocrine malfunction, causing precocious puberty or menstrual abnormalities [1,3,4,158]. Staging laparotomy with unilateral salpingo-oophorectomy followed by postoperative chemotherapy is the treatment of choice.

On pathology, the tumor manifests as a unilateral, large, predominantly solid, variegated mass. Extensive hemorrhagic and necrotic areas with cystic components containing mucoid origin are often seen.

Embryonal carcinoma resembles other MOGCTs on imaging, and findings are often nonspecific, including a large, predominantly solid mass, with extensive hemorrhagic and necrotic components and cystic areas of mucoid content [1,3,4,166].

##### Polyembryoma

Polyembryomas are extremely rare MOGCTs, typically occurring as a component of mixed GCTs. They are often associated with immature teratoma and yolk sac tumor elements and exhibit similar imaging characteristics [1,158].

##### Mixed Germ Cell Tumors

Mixed GCTs are defined by the presence of at least two different malignant germ cell components, usually a combination of dysgerminoma, immature teratomas, and yolk sac tumors [1,158]. They typically appear as large, mostly solid masses with areas of hemorrhage and necrosis. Imaging findings depend on the different histological components within the tumor [5,166].

##### Gonadoblastoma

Gonadoblastoma is a rare benign tumor with high malignant potential, consisting of a mixture of germ cells and undifferentiated sex cord cells. More than half of the cases occur in phenotypically female patients with disorders of sex development and Y chromosome material, most commonly in individuals with a Turner syndrome variant (45,X/46,XY mosaicism). Additionally, individuals with WT1-related disorders such as Frasier syndrome and Denys–Drash syndrome—both of which involve 46,XY karyotype and gonadal dysgenesis—are also at risk of developing gonadoblastoma [1,159].

Diagnosis is often made during the investigation of abnormal sex development, usually due to ambiguous external genitalia in infants. Patients may also present with precocious puberty or virilization due to hormonal activity of the tumor. Their overall prognosis is good; however, they may coexist with malignant germ cell subtypes, most commonly dysgerminoma [158].

Imaging findings are often nonspecific and poorly described in the literature. These tumors can be small and therefore difficult to detect. When large, they typically appear as solid masses, often containing mottled or punctate calcifications. Bilateral involvement is observed in up to half of the cases [1,168,169,170].

Table 1 presents clinical characteristics, laboratory findings, and imaging features that help in the characterization of pediatric ovarian germ cell tumors.

### 4.2. Sex Cord–Stromal Tumors

Sex cord–stromal tumors are a heterogeneous group of ovarian neoplasms that originate from sex cord cells, including granulosa and Sertoli cells and stromal cells, including fibroblasts, theca cells, and Leydig cells [1,3]. These tumors account for approximately 10–20% of all pediatric ovarian neoplasms. They include both benign and malignant tumors and often exhibit mixed cellular composition, leading to their classification into three categories: pure sex cord tumors, pure stromal tumors, and mixed sex cord–stromal tumors [1,3,93,171].

Sex cord–stromal tumors occur across a wide age spectrum, but their histologic subtypes demonstrate marked age-related prevalence. In the pediatric population, juvenile granulosa cell tumors (JGCTs) and Sertoli-Leydig cell tumors (SLCTs) are the most frequently encountered histologic types, corresponding approximately to 75% and 15% of SCSTs, respectively. In contrast, the most common SCSTs in adult women are fibromas and thecomas, with fibromas being more common [1,3,93,171].

Hormonal secretion is a hallmark of many sex cord–stromal tumors. Juvenile granulosa cell tumors and thecomas predominantly produce estrogens, whereas Sertoli–Leydig cell tumors are frequently associated with androgen production. Imaging plays a crucial role in the detection, characterization, and staging of SCSTs, particularly for accurately tailoring surgical planning and assessing malignant potential [1,3,93,171].

#### 4.2.1. Juvenile Granulosa Cell Tumors

Juvenile granulosa cell tumors are a malignant pure sex cord tumor. Although they account for only 5% of all granulosa cell tumors, JGCTs represent the majority of SCSTs in children and adolescents, with a mean age at diagnosis of 13 years [1,3,6,93,171]. These tumors are bilateral in 4–5% of cases.

Juvenile granulosa cell tumors are often hormonally active and typically produce estradiol, which can be used as a tumor marker, both for diagnosis and follow-up. In premenarchal girls, the excess estrogen can lead to signs of isosexual peripheral precocious puberty, including breast development, vaginal bleeding, accelerated growth, and the appearance of pubic and axillary hair. In postmenarchal patients, hormonal effects may present with menstrual irregularities, such as menorrhagia or amenorrhea. Rarely, androgen production by the tumor may lead to virilization. Juvenile granulosa cell tumors can also secrete inhibin, particularly inhibin B and Müllerian Inhibiting Substance, which can also be used as tumor markers.

Although a definitive link between enchondromatosis syndromes and JGCTs has not been confirmed, there are reports of these tumors occurring in patients with Maffucci syndrome and Ollier disease [1,3,6,159].

Despite their malignant potential, JGCTs are most often diagnosed at an early stage and have an excellent prognosis, with a survival rate of more than 90% after surgical resection. The mitotic index significantly correlates with prognosis. Long-term follow-up is essential due to the risk of late, albeit rare, recurrence [1,3,6].

Regarding pathology, JGCTs typically present as large, solid masses containing cystic spaces, with an average diameter of 12.5 cm. They are composed of primitive granulosa cells organized in both solid and follicular patterns. An intratumoral hemorrhage, necrosis, and fibrosis are commonly observed.

Imaging findings are nonspecific (Figure 18). Ultrasound and CT typically show a large, multicystic mass with solid, heterogeneous, highly vascular components and septations [93,171,172,173,174]. MRI may reveal a distinctive sponge-like pattern, characterized by numerous small cystic spaces intermixed with solid areas. Some of the cystic spaces may contain a hemorrhage, with a high signal on T1WI and fluid–fluid levels. Additionally, there may be endometrial thickening and uterine enlargement, as a result of estrogenic stimulation [1,3,4,6,93,171,172,174,175].

**Figure 18 cancers-17-02316-f018:**
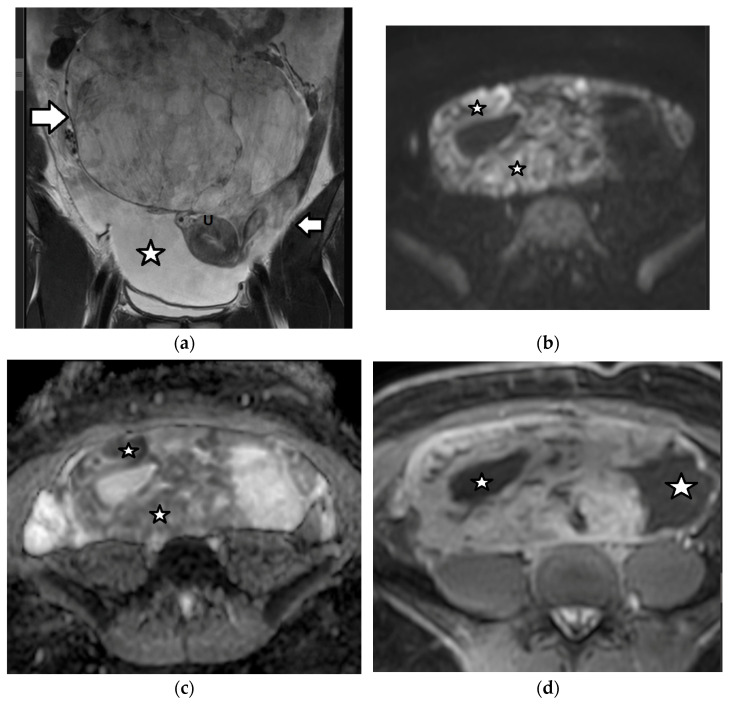
A 17-year-old girl with a right ovarian juvenile granulosa cell tumor. The patient presented with abdominal distention and a history of precocious puberty. (**a**) The coronal T2-weighted MR image of the abdomen shows a large inhomogeneous mass (arrow) originating from the right ovary and extending outside the pelvis. Ascites (star) is also seen. Note the normal left ovary (small arrow). U: uterus. The tumor exhibits diffusion restriction as shown by the hyperintense and hypointense parts (stars) of the mass on the axial (**b**) high-*b*-value diffusion-weighted image and (**c**) the corresponding apparent diffusion coefficient map, respectively. The intense contrast uptake with intratumoral cystic necrotic areas (stars) is shown on the (**d**) axial, contrast-enhanced, fat-suppressed, T1-weighted MR image.

#### 4.2.2. Sertoli–Leydig Cell Tumor

Sertoli–Leydig cell tumors are uncommon, mixed SCSTs that contain varying proportions of both Sertoli and Leydig cells. These neoplasms are classified based on the degree of differentiation into well, moderately, and poorly differentiated types. Tumors with moderate or poor differentiation are clinically malignant in approximately 10% and 60% of cases, respectively, and may exhibit retiform patterns or heterologous elements (e.g., mucinous enteric epithelium or hepatocytes). SLCTs predominantly affect females under the age of 30 years, with a median age at diagnosis of 14 years. They are unilateral in approximately 98% of cases [1,3,6].

Clinically, SLCTs are known for their androgenic effects, with virilization being the hallmark feature. Symptoms include hirsutism, deepening of the voice, acne, menstrual irregularities, or amenorrhea and clitoromegaly. Estrogenic effects may also occur, although rarely. Elevated serum AFP levels may occasionally be detected due to the presence of heterologous hepatocyte elements in poorly differentiated tumors [1,3,6].

There is a significant association between moderately and poorly differentiated SLCTs and DICER1 syndrome, a genetic condition caused by germline mutations in the *DICER1* gene, which plays a crucial role in microRNA processing. Although not definitively established, a possible association between SLCTs and Peutz–Jeghers syndrome has also been reported [1,3,6].

Prognosis is largely determined by the tumor stage at diagnosis and degree of histologic differentiation. Most cases are diagnosed at an early stage, are well or moderately differentiated, and therefore have a favorable prognosis. However, the recurrence rate may be high, especially in poorly differentiated tumors or those with heterologous elements [1,3,6,159].

The imaging findings are variable and nonspecific [1,3,6,93,171,172]. The Sertoli–Leydig cell tumor typically appears hypervascular and presents either as a predominantly solid mass, with numerous peripheral or intratumoral cysts, or as a cystic tumor, with solid components. On MRI, the fibrous stroma is characteristically identified as an area of a low signal on both T1WI and T2WI, with delayed contrast enhancement [1,3,6,93,172].

Table 2 shows the clinical findings, laboratory data, and imaging characteristics of ovarian sex cord–stromal tumors in the pediatric age cohort.

### 4.3. Epithelial Tumors

In contrast to adults, where epithelial ovarian tumors are the most common type of ovarian neoplasms, these tumors are relatively uncommon in children and adolescents, accounting for approximately 10–15% of all ovarian tumors in this age group. They typically occur after menarche, as they are associated with hormonal activity [1,3,6,123].

Epithelial ovarian tumors are categorized into benign, borderline, and malignant types. In the pediatric population, benign epithelial tumors are the most common (47–58%), borderline tumors occur less frequently (21–38%), and malignant epithelial tumors are rare (<5%) and more often low-grade and early-stage [1,3,6,123].

Although CA 125 is often elevated in epithelial ovarian malignancies in adults, its use in the pediatric population is limited and not routinely recommended, due to the low incidence of epithelial ovarian tumors in children and the lack of specificity, as this marker is often elevated in many benign conditions [1,3,6,123].

Epithelial ovarian tumors in children and adolescents pose distinct challenges in both diagnosis and management. Treatment typically involves cystectomy or oophorectomy, with an emphasis on fertility-sparing surgical approaches. While recurrence of benign epithelial tumors is uncommon, it can occur in the ipsilateral ovary. After surgery, recurrence of borderline ovarian tumors has been reported at approximately 7.7% of cases, and therefore close monitoring is recommended [1,3,6].

#### 4.3.1. Benign Epithelial Ovarian Tumors

Benign epithelial ovarian tumors include serous, mucinous, and mixed cystadenomas. Serous cystadenomas are the most common type. These tumors are typically unilateral and often present as large, cystic masses [1,3,6]. A unilocular or multilocular, homogeneous, cystic ovarian tumor filled with serous fluid and showing no enhancement after contrast administration, except for thin smooth walls or septa (<3 mm in thickness), is strongly suggestive of the diagnosis of a serous cystadenoma on CT and MRI [176]. On US, serous cystadenomas appear anechoic, with no internal vascularity [177,178].

In contrast, mucinous cystadenomas often appear as larger, multilocular cystic masses with thin walls or septa and contain mucinous fluid, which typically shows higher than water CT attenuation (>20 HU) and a high signal on T1WI due to its proteinaceous content [1,3,6,176]. Different signal intensities in the cystic locules of the tumor due to varying concentrations of mucin and hemorrhage create a characteristic “stained glass” appearance. Sonographically, these tumors contain parts with low-level internal echoes due to the presence of mucin and show no internal blood flow on Doppler examination [177,178,179].

#### 4.3.2. Borderline and Malignant Epithelial Ovarian Tumors

Imaging findings suggestive of borderline or malignant ovarian neoplasms include a large size; bilateral ovarian masses; a tumor that is partly cystic and solid with solid, vascular components, thick and irregular walls and septa, and/or papillary projections, which show the blood flow upon Doppler examination and enhancements on CT/MRI; and the presence of solid components with areas of necrosis. Ancillary findings such as pelvic organ or pelvic sidewall invasion, ascites, peritoneal metastases, and lymphadenopathy confirm the diagnosis of malignancy [1,3,6,123,176,180,181].

Borderline ovarian neoplasms are more common in the pediatric population than in older adults. Papillary projections are a characteristic feature of borderline ovarian tumors, although they may also be seen in carcinomas and, less frequently, in benign lesions. On MRI, papillary projections typically show intermediate signal intensity on T1WI, higher than that of the muscles on T2WI and contrast enhancement (Figure 19); a central hypointense fibrovascular stromal core with a hyperintense periphery on T2WI may also be seen [181,182].

Table 3 shows the clinical, laboratory, and imaging findings of pediatric epithelial ovarian tumors.

#### 4.3.3. Endometriomas

Endometriomas, the ovarian manifestation of endometriosis, are caused by the presence of endometrial glands and stroma outside the endometrial cavity. They are more common in women of reproductive age and may be unilateral or bilateral [183,184,185]. The prevalence of endometriosis in adolescence remains unknown. The incidence of the disease in young women aged ≤ 25 years presenting with chronic pelvic pain is reported to vary widely between 25% and 73%. Diagnostic imaging, including US or MRI, has a vital role in achieving an accurate diagnosis [183,184,185].

Endometriomas are cystic lesions containing viscous, dark brown fluid, which develops by the continuous accumulation of menstrual blood products of different ages, thus the term “chocolate cysts” [183]. On US, they typically appear as unilocular, thick-walled cystic masses. These cysts often have a characteristic ground-glass appearance, which is caused by low-level homogeneous echoes within the cyst, reflecting the presence of internal debris. Doppler examination reveals the absence of intralesional vascularity [186,187]. On MRI, endometriomas typically appear hyperintense on T1WI, without a loss of signal on fat saturation T1WI, due to their hemorrhagic content, and they do not enhance after gadolinium administration, as best shown on subtracted images. On T2WI, they demonstrate a gradual, homogeneous drop in signal intensity, due to the high concentration of products of repeated hemorrhage, known as the “T2 shading” sign. The hallmark imaging feature of endometriomas is the combination of hyperintensity on the T1WI and T2 shading on T2WI. This combination of findings in a cystic mass is highly suggestive of endometrioma [187,188,189]. The discrimination between hemorrhagic cysts and endometriomas may be difficult on MRI; however, most hemorrhagic cysts appear heterogeneous, less bright on T1WI, and without T2 shading [190].

## 5. Limitations

This narrative review has several inherent limitations, including the absence of a systematic search strategy, which may increase the risk of selection bias or imprecision. Additionally, the authors’ subjectivity can contribute to confirmation bias. The lack of quantitative data, formal quality assessment, and reproducibility further limit the reliability and generalizability of the findings.

## 6. Conclusions and Future Directions

Ovarian masses are rare in the pediatric population, although most abdominal masses in children and adolescents originate from the ovaries. The incidence, histologic types, and clinical features of these masses vary considerably from those seen in adults.

Accurate characterization of ovarian masses in the pediatric age cohort is crucial, as it facilitates timely referral to specialized centers for a multidisciplinary approach. This may allow ovarian-sparing procedures using minimally invasive surgical techniques, with an emphasis on fertility preservation.

Distinctive imaging findings, when combined with elevated serum tumor markers and unique clinical features, can aid in narrowing the differential diagnosis of ovarian masses in the pediatric population. Multimodality imaging, including US, MRI, or CT, plays a vital role in lesion characterization, tumor staging, and patient follow-up.

While imaging-based scoring systems have significantly improved the evaluation of ovarian masses in adults, pediatric patients require specialized models that account for their distinct disease characteristics. Future research should aim to validate and adapt these tools to enhance diagnostic accuracy, minimize unnecessary surgical interventions, and safeguard reproductive potential in the pediatric population.

Artificial intelligence (AI) has emerged as a powerful tool for extracting high-throughput information from diverse sources, including medical images. In adult gynecological imaging, AI has already proven to be a valuable asset. Extending the application of AI-based tools to the evaluation of pediatric ovarian masses holds significant promise. These technologies can aid in lesion characterization, risk stratification, and prognostic assessment, ultimately supporting more accurate and efficient clinical decision-making.

## Figures and Tables

**Figure 1 cancers-17-02316-f001:**
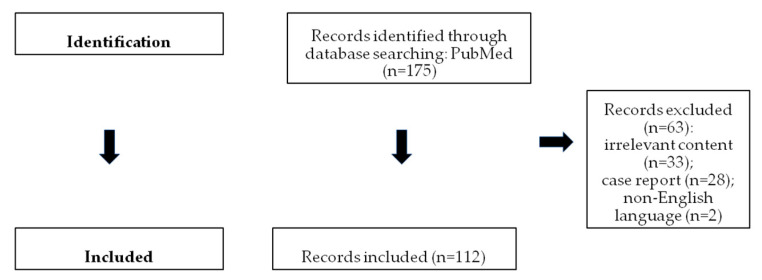
Flow chart showing study selection.

**Figure 2 cancers-17-02316-f002:**
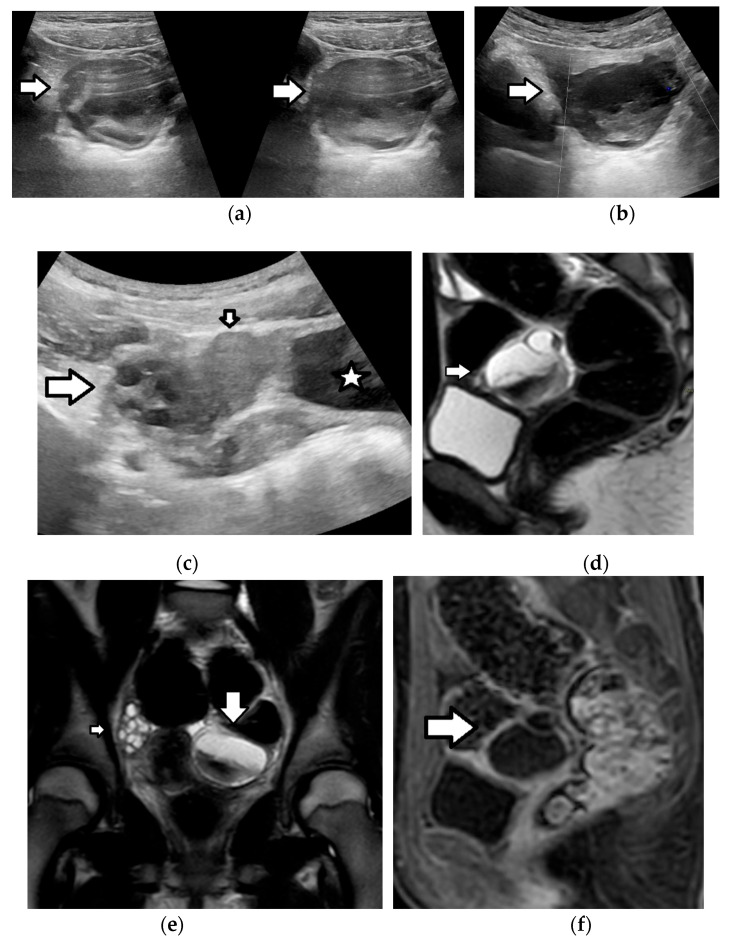
A 13-year-old girl with back pain and a history of recent abdominal trauma. Transverse (**a**) grayscale and (**b**) color Doppler US images depict a well-defined complex left ovarian mass (arrow), with cystic parts, internal echoes, and an absence of vascularity. (**c**) The transverse grayscale US image shows a normal right ovary (arrow), with ellipsoid morphology and small cystic follicles. The uterus (small arrow). A left ovarian mass lesion (star). The T2-weighted MR images in a (**d**) sagittal and (**e**) coronal plane demonstrate a multicystic left ovarian lesion (arrow), with fluid–fluid levels and markedly hypointense dependent parts, due to the presence of a hemorrhage. The normal right ovary (small arrow) is also seen. (**f**) The sagittal, subtracted, dynamic-contrast-enhanced, T1-weighted MR image shows peripheral, smooth, thin-lesion enhancement (arrow), with the absence of nodular projections or solid components. Imaging features were suggestive of a hemorrhagic cyst. Sonographic follow-up revealed lesion resolution.

**Figure 3 cancers-17-02316-f003:**
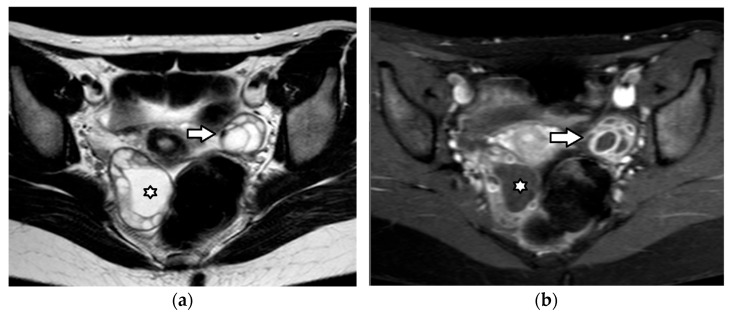
A 15-year-old girl with normal ovaries. The axial (**a**) T2-weighted and (**b**) fat-suppressed, contrast-enhanced, T1-weighted MR images demonstrate normal-sized ovaries with multiple small cystic follicles. In the right ovary, a dominant follicle (star) appears as a unilocular, thin-walled, non-enhancing cystic lesion. In the left ovary, a small cystic lesion with a slightly thickened, contrast-enhancing wall (arrow) is seen, consistent with a corpus luteum.

**Figure 4 cancers-17-02316-f004:**
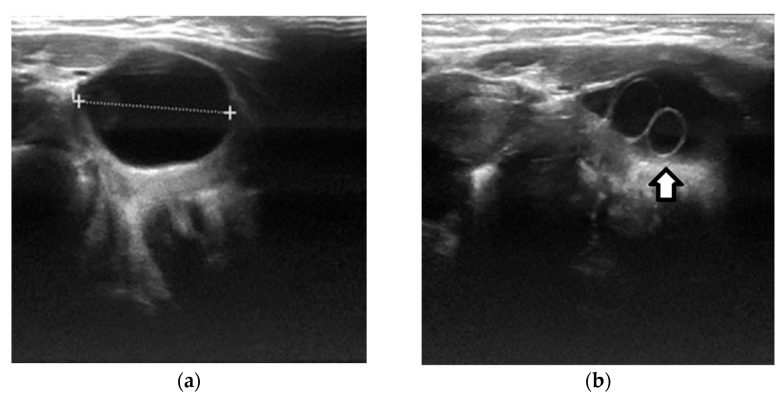
Left follicular cyst in a 31-week preterm infant, detected incidentally on US. (**a**,**b**) The transverse grayscale US images show an anechoic ovarian lesion (cursors) with a smooth, thin-wall, and small daughter cyst (arrow). Follow-up US performed after eight weeks demonstrated the resolution of the lesion.

**Figure 5 cancers-17-02316-f005:**
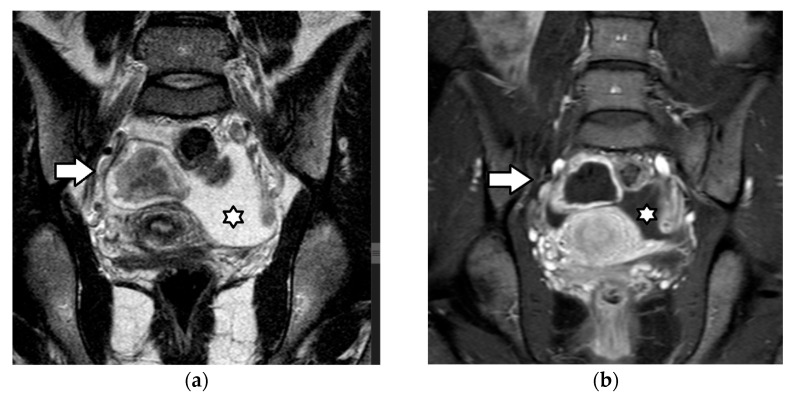
Corpus luteum cyst rupture in a 15-year-old girl presenting with acute abdominal pain. A coronal (**a**) T2-weighted MR image and a (**b**) contrast-enhanced, T1-weighted image with fat suppression demonstrate a hemorrhagic cystic lesion in the right ovary (arrow), characterized by an irregular, thickened, enhancing wall, findings typical of a corpus luteum cyst. A small amount of free fluid (star) is also observed in the pelvic cavity.

**Figure 6 cancers-17-02316-f006:**
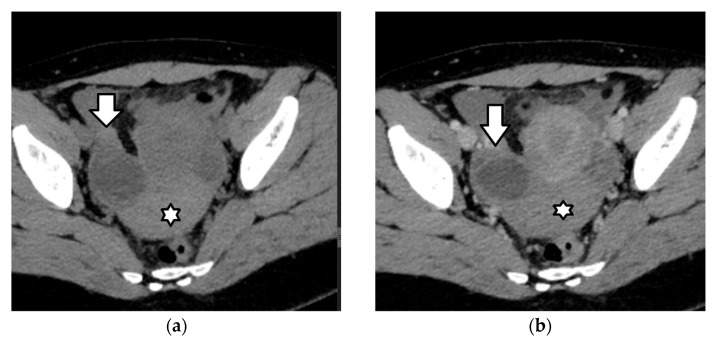
Corpus luteum cyst rupture and hemoperitoneum in a 15-year-old girl. The axial CT images of the pelvis (**a**) before and (**b**) after intravenous contrast medium administration demonstrate a thick-walled cystic lesion in the right ovary (arrow), consistent with a corpus luteum cyst. The associated hyperdense fluid in the pelvis (star) represents hemoperitoneum.

**Figure 7 cancers-17-02316-f007:**
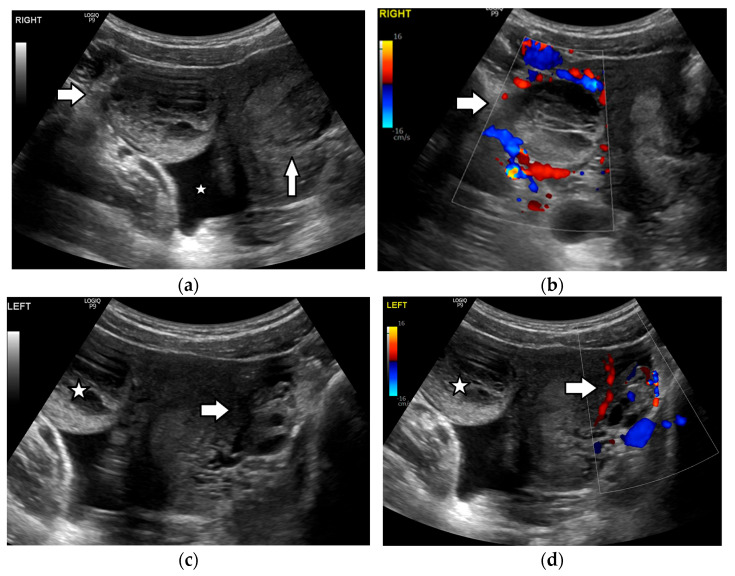
Right hemorrhagic ovarian cyst in a 15-year-old girl, with acute pelvic pain. The transverse (**a**) grayscale and (**b**) color Doppler US images show a complex cystic right ovarian mass (arrow), with lace-like echoes, an absence of internal blood flow, and peripheral vascularity. A moderate amount of fluid in the Douglas space (star (**a**)) is seen. The uterus (long arrow (**a**)). The transverse (**c**) grayscale and (**d**) color Doppler US images demonstrate a normal left ovary (arrow). A right hemorrhagic ovarian cyst (star).

**Figure 8 cancers-17-02316-f008:**
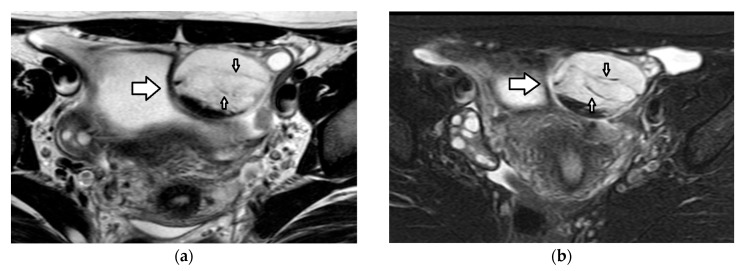
Left hemorrhagic ovarian cyst in a 16-year-old girl. The axial (**a**) T2-weighted and (**b**) fat-suppressed, T2-weighted MR images demonstrate a predominantly hyperintense cystic mass in the left ovary (arrow). Internal fibrin strands are visible (small arrows), producing the characteristic “fishnet” appearance, which is typical of hemorrhagic ovarian cysts.

**Figure 9 cancers-17-02316-f009:**
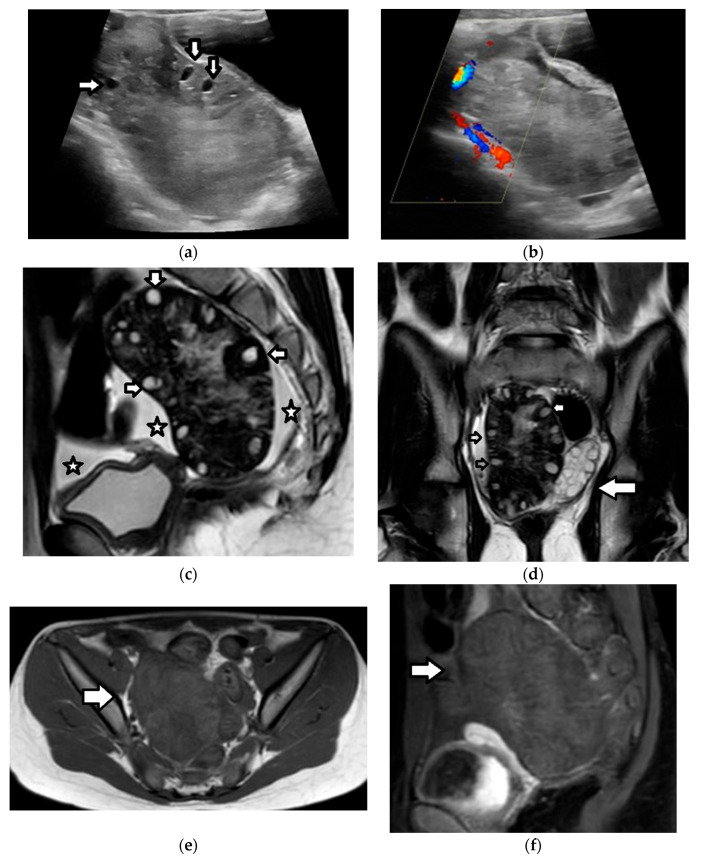
Right ovarian torsion in an 8-year-old girl, presenting with acute lower quadrant pain following recent minor blunt abdominal trauma. (**a**) The sagittal grayscale US image depicts prominent peripheral follicles (arrows) as small cystic structures in an enlarged, heterogeneous right ovary. The volume of the right ovary is 91.30 cm^3^. (**b**) The sagittal color Doppler US image shows the absence of blood flow in the right ovary, consistent with ovarian torsion. The T2-weighted MR images in the (**c**) sagittal and (**d**) coronal plane show an enlarged, inhomogeneous right ovary, with edematous stroma and multiple, peripherally located follicles (small arrows). A small amount of ascites (star (**c**)) is seen. A normal left ovary (arrow (**d**)), with an ellipsoid shape and hyperintense follicles randomly distributed throughout the parenchyma. (**e**) The axial, T1-weighted MR image demonstrates an enlarged right ovary (arrow), with a slightly hyperintense appearance, due to the presence of a hemorrhage. (**f**) The sagittal, fat-suppressed, contrast-enhanced, T1-weighted image depicts the absence of ovarian enhancement (arrow). Ovarian torsion was confirmed at surgery.

**Figure 10 cancers-17-02316-f010:**
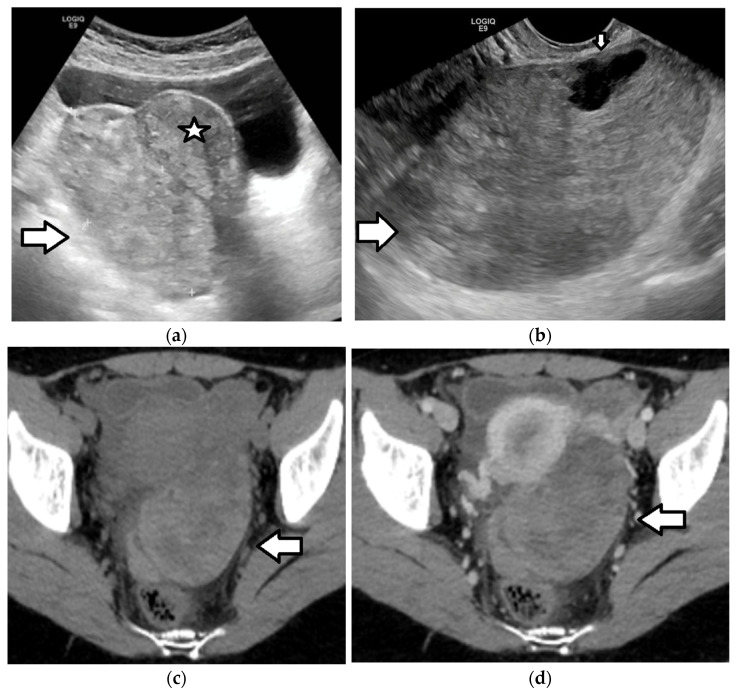
A 19-year-old woman with ovarian torsion. A large predominantly echogenic left adnexal mass (arrow) is shown on (**a**) transabdominal and (**b**) transvaginal US, indicative of an enlarged ovary. Note the presence of a follicle (small arrow) at the periphery of the ovary surrounded by an echogenic (edematous) rim. Normal uterus (star). (**c**) An axial unenhanced CT image of the same patient demonstrates diffuse hyperdensity of the ovarian parenchyma (arrow), consistent with a hemorrhage. No parenchymal enhancement (arrow) was detected on the corresponding post-contrast CT (**d**).

**Figure 11 cancers-17-02316-f011:**
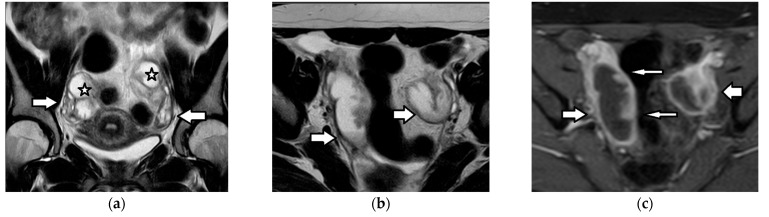
A 19-year-old woman with bilateral pyosalpinx. T2-weighted MR images in (**a**) coronal and (**b**) axial plane show normal ovaries (arrows) and dilated tubes bilaterally (stars). Note the fluid–fluid levels within the dilated tubes (arrows in (**b**)). (**c**) Axial fat-suppressed contrast-enhanced T1-weighted MR image depicts diffuse tubal wall thickening (arrows) and multiple thickened endosalpingeal folds (long arrows).

**Figure 12 cancers-17-02316-f012:**
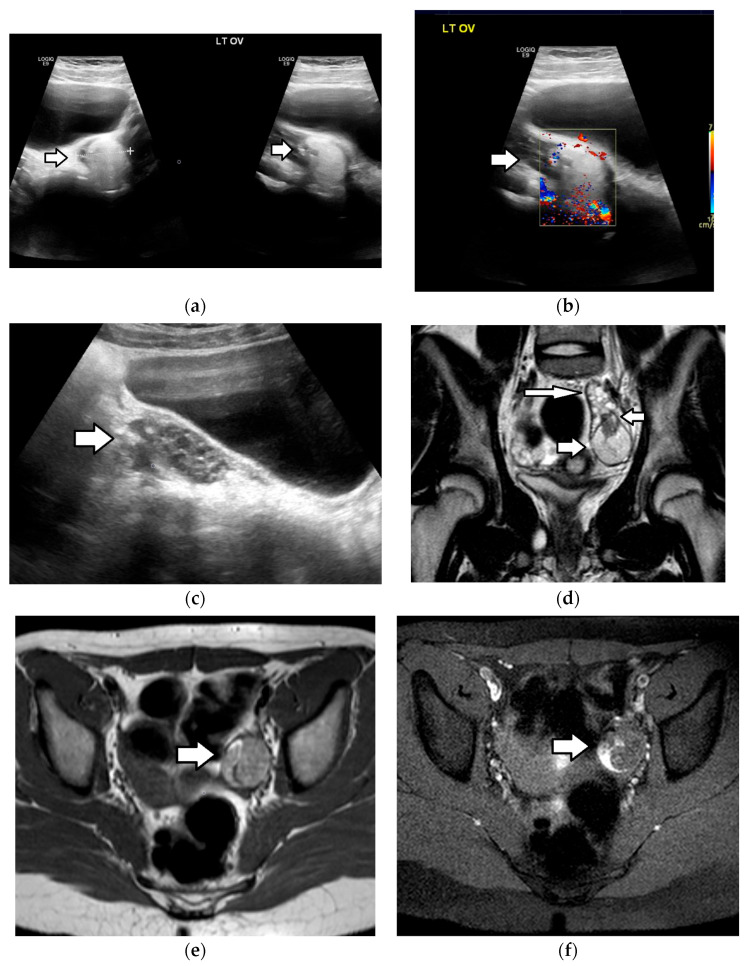
Mature teratoma of the left ovary in a 15-year-old girl incidentally found on sonography. (**a**) Grayscale US images depict a heterogeneous, mainly hyperechoic left ovarian mass (arrow). (**b**) Color Doppler US reveals the absence of internal vascularity (arrow). (**c**) The grayscale US image shows the right normal ovary (arrow). (**d**) The coronal T2-weighted MR image depicts a well-demarked, heterogeneous mass lesion (arrow) originating from the left ovary (long arrow). The tumor appears mainly hyperintense, with a nodule of the intermediate signal (Rokitansky nodule, small arrow). Axial (**e**) T1-weighted and (**f**) fat-saturated T1-weighted MR images. The mass (arrow) appears inhomogeneous, predominantly with a high T1 signal and signal drop on a fat-suppressed image, a finding indicating the presence of fat and, therefore, the diagnosis of a mature teratoma.

**Figure 13 cancers-17-02316-f013:**
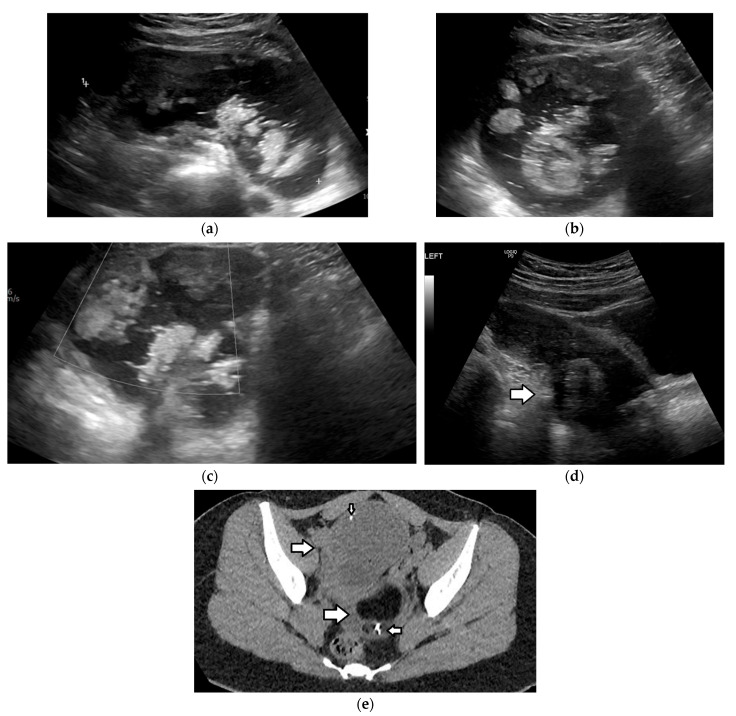
Bilateral mature teratomas in a 15-year-old girl presenting with abdominal pain. The grayscale US images in a (**a**) sagittal and (**b**) transverse orientation demonstrate a large heterogeneous, cystic, unilocular right adnexal lesion (cursors (**a**)). The tumor has smooth contours and contains hyperechoic elements and echogenic linear interfaces floating within the cyst, representing hair fibers (“dermoid mesh” sign). (**c**) Color Doppler US shows no intratumoral vascularity. (**d**) A sagittal grayscale US image of the left adnexa shows a heterogeneous, mainly hyperechoic mass (arrow) that contains areas of acoustic shadowing. The sonographic findings are suggestive of bilateral mature cystic teratomas. (**e**) The axial unenhanced CT image of the same patient depicts a heterogeneous, predominantly fatty left adnexal tumor (arrow) with small calcifications (small arrow), compatible with the diagnosis of a mature teratoma. The right adnexal tumor (arrow) appears mainly cystic, with tiny calcifications (small arrow).

**Figure 14 cancers-17-02316-f014:**
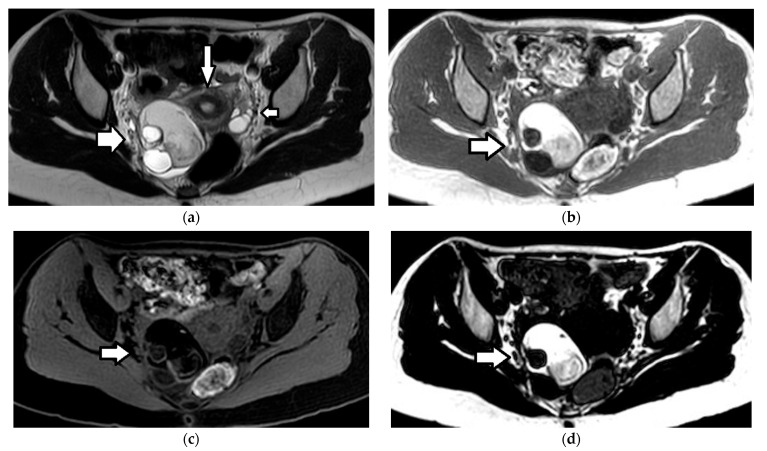
Right ovarian mature teratoma in a 19-year-old girl. (**a**) Axial T2-weighted MR image. DIXON (**b**) T1-weighted, (**c**) fat-suppressed, and (**d**) fat-only MR images. A well-delineated, heterogeneous, multicystic right ovarian tumor (arrow) is detected. A large part of the mass has an intermediate T2 signal and high T1 signal, similar to that of the subcutaneous fat. Suppression of the high signal is observed on the fat-saturated, T1-weighted MR image, and hyperintensity of the same part is seen on the fat–water separation sequence, suggestive of the presence of macroscopic fat. MRI findings are typical of a mature cystic teratoma. A normal left ovary (small arrow (**a**)) and uterus (long arrow (**a**)).

**Figure 15 cancers-17-02316-f015:**
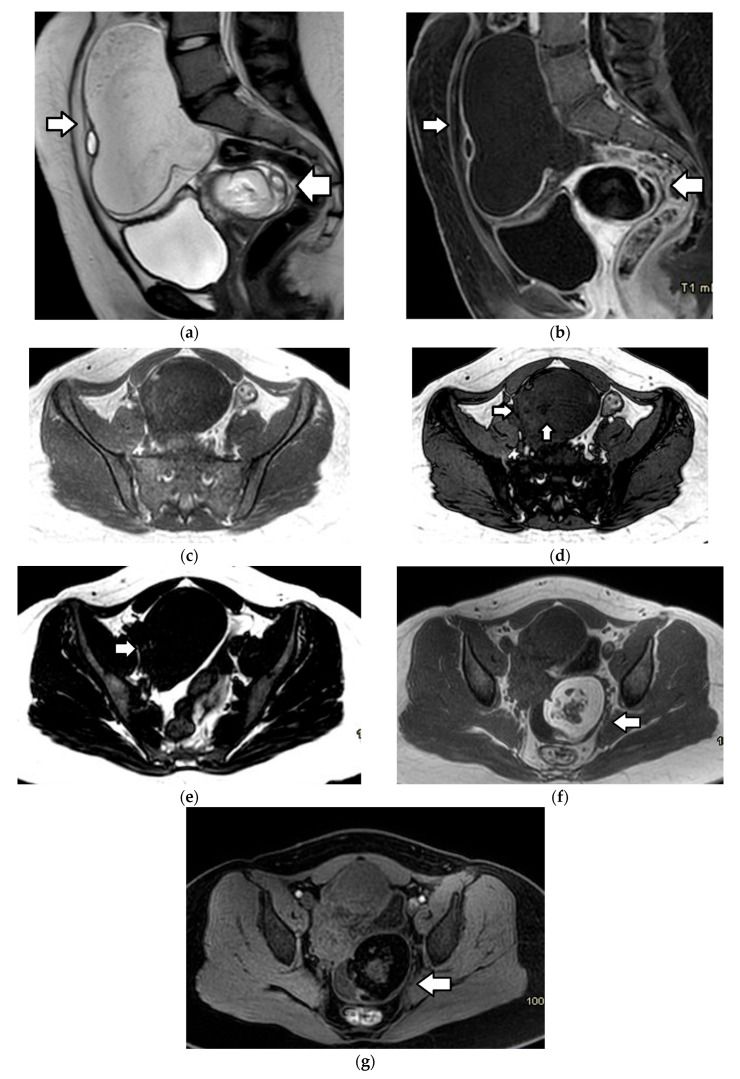
Bilateral mature teratomas in a 15-year-old girl (same patient as in Figure 13). (**a**) The sagittal, T2-weighted MR image shows bilateral ovarian masses (arrows) that are mainly hyperintense. (**b**) The sagittal, fat-saturated, contrast-enhanced, T1-weighted MR image depicts the absence of solid contrast-enhancing intratumoral components (arrows). Axial DIXON (**c**) T1-weighted, (**d**) opposed-phase, and (**e**) fat-only MR images. Small areas of a signal drop on opposed-phase imaging and/or hyperintensity on fat-only imaging (small arrows) denote the presence of microscopic fat within the ovarian mass. Axial (**f**) T1-weighted and (**g**) fat-saturated T1-weighted MR images depict a heterogeneous left ovarian tumor (arrow), with a large component detected with a high T1 signal and drop of the signal on fat-suppressed, T1-weighted imaging due to the presence of fat. The imaging findings are suggestive of bilateral mature teratomas.

**Figure 16 cancers-17-02316-f016:**
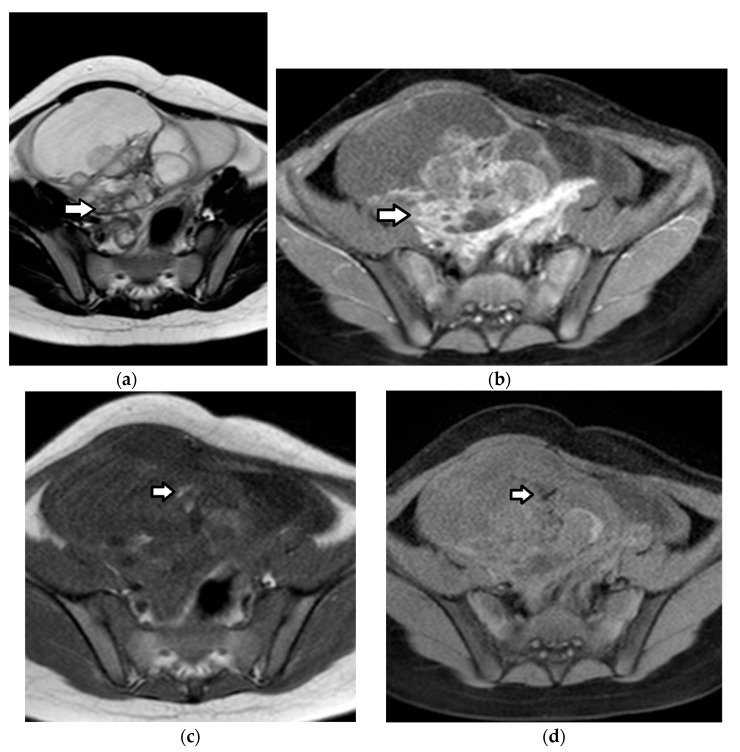
Immature teratoma of the right ovary in a 7-year-old girl presenting with abdominal distention and vomiting. The axial (**a**) T2-weighted and (**b**) fat-suppressed, contrast-enhanced, T1-weighted images demonstrate a large, multicystic right adnexal mass, with solid components (arrow) enhancing after contrast material administration. The axial (**c**) T1 and (**d**) fat-saturated, T1-weighted images demonstrate small hyperintense areas (small arrow) within the tumor, with signal loss on images with fat saturation, compatible with fat droplets.

**Figure 17 cancers-17-02316-f017:**
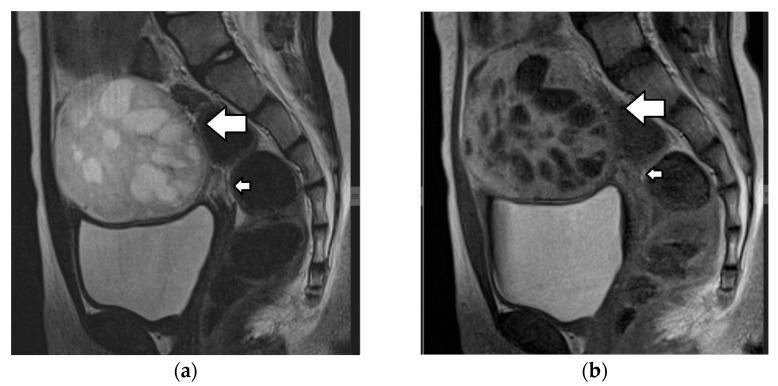
A 6-year-old girl with an ovarian yolk sac tumor. The sagittal (**a**) T2-weighted and (**b**) post-contrast, T1-weighted MR images show a predominantly solid tumor with multiple cystic areas within the pelvis (arrow). Prepubertal uterus (small arrow).

**Figure 19 cancers-17-02316-f019:**
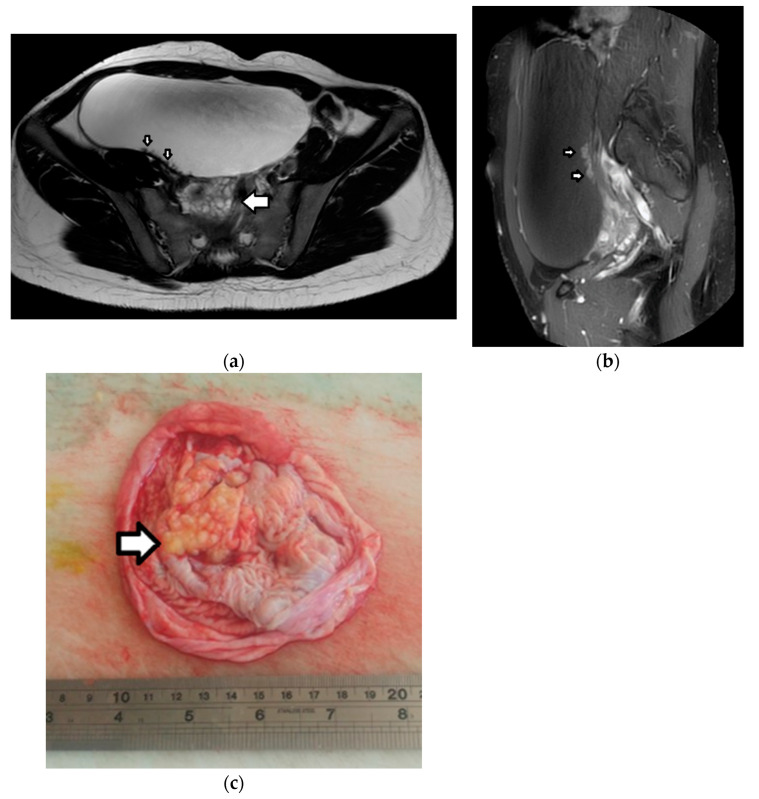
An 18-year-old woman with right serous borderline cystadenoma. (**a**) The axial T2-weighted MR image demonstrates a large cystic tumor originating from the right adnexa with multiple small papillary projections at its periphery (small arrows), which show intense enhancement on the (**b**) sagittal, post-contrast, fat-suppressed, T1-weighted image. Note the displaced normal left ovary (arrow (**a**)). (**c**) Macroscopic examination of the surgical specimen confirmed the presence of a solid component in the lesion’s wall (arrow).

**Table 1 cancers-17-02316-t001:** Clinical features, laboratory results, and imaging findings that help in the characterization of ovarian germ cell tumors in the pediatric population (GCT: germ cell tumor; MOGCTs: malignant ovarian germ cell tumors; MCT: mature cystic teratoma; AFP: alpha fetoprotein; LDH: lactate dehydrogenase; β-HCG: beta-human chorionic gonadotropin).

Histology	Clinical Features	Biologic Behavior	Laboratory Findings	Imaging Characteristics
**Mature teratoma**	**most common benign ovarian tumor** most common GCT mean age: first, second decades often asymptomatic bilateral: 10–25% 10% of MOGCTs, in contralateral ovary	benign complications: torsion, rupture, infection, malignant transformation (rare)		mean diameter: 6.5 cm **US:** • **unilocular cystic mass + hyperechoic nodule**, with echogenic dots/strands or posterior acoustic shadowing • no internal vascularity • “tip of the iceberg” sign • fat–fluid levels • “dermoid mesh” sign • echogenic mass **CT: complimentary, highly diagnostic** • **large amounts of fat and/or coarse calcifications** • fat–fluid levels • “floating balls” sign • “Poké Ball” sign **MRI: complimentary, highly diagnostic** • **macroscopic fat: high T1 and T2 signal, suppression at fat saturation sequences** • fat–fluid levels • microscopic fat: signal drop on opposed phase T1WI or high signal on fat only DIXON T1WI • keratin: diffusion restriction • calcifications: markedly hypointense
**Immature teratoma**	10–20% of ovarian malignancies second most common MOGCT mean age: 10 years ipsilateral MCT: 26% contralateral MCT: 10%	more aggressive compared to MCT often diagnosed at an early stage favorable prognosis	**increased AFP** (yolk sac tumor elements): 33–65%	**large size** (mean diameter: 16 cm) extensive solid parts with rich vascularity small cystic areas scattered foci of fat multiple, small, irregular calcifications
**Dysgerminoma**	**most common ovarian malignancy** 30% of MOGCTs mean age: second, third decades; 10%: first decade bilateral: 10–15% **associated with gonadal dysgenesis, gonadoblastoma, chromosomal abnormalities, e.g., Turner syndrome**	often diagnosed at an early stage favorable prognosis involvement of pelvic or retroperitoneal lymph nodes recurrence: 13–20%	**increased LDH: 95%** increased β-hCG (syncytiotrophoblastic giant cells): 5%	large, well-defined, lobulated, mainly solid tumor **fibrovascular septa: characteristic** • US: hypoechoic, vascular • MRI: low T2 signal, strong enhancement areas of necrosis, hemorrhage, cystic changes, and calcifications
**Yolk sac tumor**	second most common MOGCT mean age: second, third decades	rapid growth aggressive often diagnosed at an early stage	**increased AFP**	large, predominantly solid tumor, highly vascular or solid and cystic mass, with intratumoral hemorrhage and necrosis **MRI** • **“bright dot” sign:** signal voids on unenhanced imaging or contrast-enhancing vessels (dilated vessels or aneurysms): **typical finding**
**Non-gestational choriocarcinoma**	very rare MOGCT often part of mixed GCT	aggressive poor prognosis	**increased β-hCG**	few data complex, well-defined, highly vascular tumor
**Embryonal carcinoma **	rare MOGCT often part of mixed GCT mean age: 14 years	highly malignant	**increased AFP or ** **β-hCG:** 60%	**nonspecific** large, predominantly solid mass, extensive hemorrhagic and necrotic areas, cystic components with mucoid content
**Polyembryoma**	extremely rare MOGCT often part of mixed GCT			**nonspecific**
**Mixed GCT**	often a combination of dysgerminoma, immature teratoma, and yolk sac tumor			**nonspecific** large, predominantly solid tumor areas of hemorrhage and necrosis
**Gonadoblastoma**	rare **association with disorders of sex development and Y chromosome material** (Turner syndrome variant) + **WT1-related disorders** such as Frasier syndrome and Denys–Drash syndrome bilateral: 50% may coexist with MOGCTs, often dysgerminoma	good prognosis		**nonspecific** small size: difficult to detect or large, solid tumor, with mottled or punctate calcifications

**Table 2 cancers-17-02316-t002:** Clinical features, laboratory results, and imaging findings that help in the characterization of ovarian sex cord–stromal tumors in a pediatric population (SCSTs: sex cord–stromal tumors; AFP: alpha fetoprotein).

Histology	Clinical Features	Biologic Behavior	Laboratory Findings	Imaging Characteristics
**Juvenile granulosa cell tumor**	75% of SCSTs mean age: 13 years bilateral: 4–5% **secretes estrogens: signs of isosexual peripheral precocious puberty or menstrual irregularities** secretes androgens: virilization (rarely) **possible association with enchondromatosis syndromes: Maffuci syndrome, Ollier disease**	malignant often diagnosed at an early stage excellent prognosis recurrence: rare	**estrogens** androgens (rarely) **inhibin**, especially inhibin B Müllerian Inhibiting Substance	**nonspecific** large size (mean diameter, 12.5 cm) multicystic tumor, with solid vascular components **MRI** • **sponge-like pattern:** numerous small cystic spaces (may contain hemorrhage: high T1 signal and fluid–fluid levels) and solid components • **endometrial thickening and uterine enlargement**
**Sertoli–Leydig cell tumor**	15% of SCSTs mean age: 14 years mostly unilateral **secretes androgens: virilization** secretes estrogens (rarely) **association with DICER1 syndrome**(moderately and poorly differentiated tumors) **possible association with Peutz–Jeghers syndrome**	moderately and poorly differentiated tumors: clinically malignant in 10% and 60% of cases, respectively often diagnosed at an early stage favorable prognosis recurrence rate: may be high	**increased AFP** (occasionally): poorly differentiated tumors with heterologous hepatocyte elements	**nonspecific** predominantly solid vascular tumor, with numerous peripheral or intratumoral cysts or cystic, with solid components **MRI** • **fibrous stroma:** low T1 and T2 signal and delayed enhancement

**Table 3 cancers-17-02316-t003:** Clinical features, laboratory results, and imaging findings that help in the characterization of epithelial ovarian tumors in the pediatric age group.

Histology	Clinical Features	Biologic Behavior	Laboratory Findings	Imaging Characteristics
**Benign epithelial tumors: serous, mucinous, and mixed cystadenoma**	Most common (47–58%) of epithelial tumors After menarche	Benign Recurrence: rarely in contralateral ovary		**Serous cystadenoma:** unilocular or multilocular, homogeneous, cystic tumor with serous fluid, absence of internal vascularity on Doppler US, and contrast enhancement on CT/MRI, except for thin, smooth walls or septa (≤ 3 mm) **Mucinous cystadenoma:**multilocular cystic tumor, large size, absence of internal vascularity on Doppler US, and contrast enhancement on CT/MRI, except for thin, smooth wall or septa (≤3 mm), and presence of mucinous fluid, with higher than water CT density (>20 HU) and high T1 signal (proteinaceous content); **“stained glass” appearance:**different signal intensity in the cystic parts of the tumor due to variable concentrations of mucin and hemorrhage as a **characteristic finding**
**Borderline and malignant epithelial tumors**	**Borderline:** 21–38% of epithelial tumors, and more common in children and young women compared to in adults **Malignant:** rare (<5%)	**Borderline:** may recur (7.7%) **Malignant**: early-stage and low-grade		Large size, bilateral masses, cystic-solid tumor, with solid, vascular components, thick and irregular walls and septa, and/or papillary projections, with blood flow on Doppler US and enhancement on CT/MRI, as well as solid tumor with areas of necrosis Ancillary findings: pelvic organ or pelvic sidewall invasion, ascites, peritoneal metastases **Papillary projections:** intermediate T1 and T2 signal, variable vascularity, and may have a central hypointense fibrovascular stromal core and hyperintense periphery on T2WI, which are **mostly characteristic of borderline tumors**

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
