# Peer review of "Imaging Evaluation of Ovarian Masses in a Pediatric Population: A Comprehensive Overview"

_cancers, 2025, doi:10.3390/cancers17142316_

Round 1

Reviewer 1 Report

Comments and Suggestions for Authors

General remarks

The topic of the review is not new and multiple similar articles have been published to date. For instance, a recent review by Birbas et al. cited as Ref.#3 that has been published in MDPI’s journal “Children” in 2023 contains very similar data and statements. Whilst I agree with the Authors that the ideal approach to diagnostic imaging ovarian masses has not been determined, I feel that their review has some serious omissions which undermine its ability to guide clinicians. The study design is not clearly stated, in particular there is no information on the scientific background and rationale for the investigation. Key elements of the study design are not presented in the Abstract and in the Introduction of the manuscript. In particular, the study setting, period of data collection for this review were not clearly stated. The inclusion criteria for this review are not presented and the Authors have not included a flow chart containing all potential retrieved citations. In general, reviews like the presented one are heavily influenced by the inclusion criteria used and so are also open to bias as with any other approach to reviewing the scientific literature. The manuscript lacks a discussion on the potential limitations of the study, taking into account sources of potential bias or imprecision.

Detailed opinion

First, the studied populations were not clearly defined. The Authors state that their aim was to describe adnexal masses “in children and in young women” with no specification what did they mean when they used the term “young women”. It is unclear what was the age limit for such a group-was 18 years or maybe 25 years?

The Authors present multiple imaging data of adnexal mases and among others include two lesions in two young women that can hardly be compared with adolescent girl’s population and would easily fit the “adult” women group. Figure 4 (line 158) presents a ruptured corpus luteum cyst in a 22 years old woman and figure 18 (line 552) presents a case of immature teratoma of the ovary in a 23-year-old woman. It is surprising that nowhere in this review a clear differentiation of the various age groups of pediatric and adolescent (PAG) patients with adnexal tumors can be found. These groups are particularly important in adnexal masses ultrasound imaging when the frequency of various benign and malignant tumors varies substantially. In the presented reference of the study of Oltmann et al. [2010] the investigators have found that female patients aged 1 to 8 years have the greatest incidence of ovarian malignancy. The Author’s statement on reported frequency of adnexal malignancies mentioned as 10% up to 30% in paediatric and adolescent patients seems extremely exaggerated. For instance, Lam et al [ 2018] and Heo et al. [2014] have reported that ovarian malignancy could be found in 3–8% of children and adolescents diagnosed with an adnexal mass. Again, the Author’s statement “Up to 65% of ovarian masses in this age population are neoplastic” is hard to understand.

In the Introduction section the Authors have also cited an annual rate of 2,6 cases per 100.000 patients. However, it is not clear what did they really mean by citing this figure. This piece of information did not refer to a specific time, population and/or country. Moreover, a close look at the cited supporting references numbered 1-9 reveals that none of them has been based on large epidemiologic studies. The Authors should consider discussing examples of the relevant population-based retrospective cohort studies that would analyse the age groups of girls ≤18 years with ovarian/adnexal masses only. Moreover, to make the review meaningful, the discrimination of such masses should rather be addressed in separate groups of patients, for example in purely pediatric population, like in neonates, and in prepubertal and postpubertal patients. The prevalence of ovarian masses in these groups could then be meaningfully compared to the frequency of ovarian lesions found in the  “young” women category. However, no such information is presented in the reviewed manuscript.

Second, I do not understand why the Authors have chosen to present only a median tumor size for various groups of adnexal masses. The median size without range assessment of various types of tumors is virtually useless in a single case of ovarian lesion diagnostics. The Authors should consider to add information on the tumor size range in mm, or like in the case of the medians, also add upper and lower quartiles for each group of ovarian lesions. For specific types of masses, like e.g. serous and mucinous cystadenomas it would also be more valuable to present  both the median diameter and range, i.e the smallest and the largest tumor size that have been reported for each group. 

Third, the reader would certainly be interested on how the US morphology of adnexal/ovarian masses were described. In particular it would be interesting to present not only the  propoprtion of  unilateral and bilateral  lesions but also how many tumors in each group were unilocular, multilocular, unilocular-solid, multilocular-solid as well as purely solid.  Next, tumor vascularity as seen on color/power Doppler sonography should be presented. In such cases, the description should mention the existing and well-established criteria, like for instance the IOTA group COLOR score of ovarian lesion assessment. 

Fourth, it is not clear from the presented data if the Authors were aware of the data on the proportion of ovarian tumors in each particular group that before surgery were correctly classified by the original ultrasound examiner. If such information is currently not available for a paticular ovarian tumor type, at least the most typical features for each separate group in pediatric and adolescent population should be presented. By using multiple citations on the ultrasound discrimination of ovarian tumors, the Authors admit that in this respect their “young women” category does not differ from the data cited for the “adult” female population.

Fifth, the Authors should also consider to comment on the currently used methods of ovarian masses discrimination, not only on presented “pattern recognition”, i.e. expert assessment of tumor morphology, but also on other existing scoring systems like: RMI, ROMA or IOTA group models. If, to date, such information has not been available for particular age groups and specific ovarian masses , it should be clearly stated. The same problem is related to other maging studies like CT or MRI in adolescent patients.

It is not clear why the Authors firmly suggest that the radiologists should use the ORADS-US system for ovarian masses in pediatric population when they also admit that this system has not been validated in this age group. They state (lines 95-96), quote “Pediatric radiologists should apply the Ovarian-Adnexal Reporting and Data US System (O-RADS US) when assessing an ovarian lesion…”.

Although the recent important study is mentioned as a citation, there is no detailed discussion with the data presented by Cooper et al.[2024]. These Authors have attempted not only to describe normative imaging findings and adnexal masses features in pediatric and adolescent populations, but also they have compared the performance of expert opinion alone, as well as using retrospective application of the International Ovarian Tumor Analysis (IOTA) simple rules (SRs) and benign descriptors (BDs) in those found to have an adnexal mass. More importantly, their analysis was performed on three different age groups: neonatal (aged < 1 year), premenarchal (aged ≥ 1 year) and postmenarchal up to 21 years old.

Sixth, in the Introduction (lines 54-55) the Authors state that, quote: “Common non-neoplastic ovarian lesions in children and young females include functional cysts, endometriomas…” It may be controversial, but typically endometriomas are related to an abnormal growth of the ovarian tissues and because of that these lesions should be classified as a form of neoplastic growth within the ovary. Moreover, these masses, like breast or endometrial cancer, depend on estrogenic hormonal stimulation, and, although rarely, may mimic the ovarian malignancy or over the years may transform into malignancy. Taking into account these considerations, endometriomas should rather be described as benign neoplasms and not as “non-neoplastic ovarian lesions”. According to the WHO 2020 classification endometrial tumors, both benign or malignant, belong to the category “ovarian epithelial neoplasms” and certainly do not belong to “non-neoplastic lesions” (former group 9). Therefore the “endometrioma” subchapter should be placed in 3.3.1 category (starting in line 779) that describes “benign epithelial ovarian neoplasms”.

Seventh, the table that starts on page 28 and extends to page 36 is apparently too long, contains too many data and because of these reasons it is very difficult to read. A summary for each tumor type or tumor groups in separate tables would be more comprehensive.

Finally, the Author’s “Conclusions” section is a brief summary of existing textbook-type opinions and does not contain any new knowledge on the presented topic. Very similar conclusions have been already presented in older and newer systematic reviews referring to imaging and diagnostic issues in female pediatric and adolescent patients with ovarian masses. The Authors should rather consider different approach and conclude which groups of ovarian masses deserve multimodality imaging, including US, MRI or CT. It is apparent that not all, but only few adolescent patients would need such a complex and costly approach. Certainly, multicenter and larger studies are required for the prospective validation of imaging features and diagnostic models in PAGs with ovarian masses. Such new and prospectively validated strategies could aid an appropriate patient’s referral and enable to select the tailored patients care including fertility-sparing approach.

Author Response

The topic of the review is not new and multiple similar articles have been published to date. For instance, a recent review by Birbas et al. cited as Ref.#3 that has been published in MDPI’s journal “Children” in 2023 contains very similar data and statements. Whilst I agree with the Authors that the ideal approach to diagnostic imaging ovarian masses has not been determined, I feel that their review has some serious omissions which undermine its ability to guide clinicians. The study design is not clearly stated, in particular there is no information on the scientific background and rationale for the investigation. Key elements of the study design are not presented in the Abstract and in the Introduction of the manuscript. In particular, the study setting, period of data collection for this review were not clearly stated. The inclusion criteria for this review are not presented and the Authors have not included a flow chart containing all potential retrieved citations. In general, reviews like the presented one are heavily influenced by the inclusion criteria used and so are also open to bias as with any other approach to reviewing the scientific literature. The manuscript lacks a discussion on the potential limitations of the study, taking into account sources of potential bias or imprecision.

Answer: Thank you for pointing these out. We agree with these comments. Our study is a narrative review of the imaging findings of ovarian masses in the pediatric population, aiming to identify imaging features which, when combined with clinical characteristics and laboratory findings, can significantly aid in lesion characterization.
Details regarding the study design have been added to both the Abstract and the Introduction section (Lines: 39-42, and 163-171). A flowchart illustrating the study selection process has also been included as Figure 1.
Study limitations are also discussed (Lines: 936-941).

Detailed opinion

Comment 1: First, the studied populations were not clearly defined. The Authors state that their aim was to describe adnexal masses “in children and in young women” with no specification what did they mean when they used the term “young women”. It is unclear what was the age limit for such a group-was 18 years or maybe 25 years?

Response 1: Thank you for pointing this out.

The present study is a narrative review of the imaging findings of ovarian masses in the pediatric population, aiming to identify imaging features which, when combined with clinical characteristics and laboratory findings, can significantly aid in lesion characterization.

We used the term «pediatric population» in the revised version of the manuscript, based on the following definition: Pediatric malignancies” are defined as those malignancies commonly associated with the pediatric population that may arise during childhood, adolescence or young adulthood. [Long-Term Follow-Up Guidelines for Survivors of Childhood, Adolescent, and Young Adult Cancers. Version 6.0 (October 2023)].

Comment 2: The Authors present multiple imaging data of adnexal mases and among others include two lesions in two young women that can hardly be compared with adolescent girl’s population and would easily fit the “adult” women group. Figure 4 (line 158) presents a ruptured corpus luteum cyst in a 22 years old woman and figure 18 (line 552) presents a case of immature teratoma of the ovary in a 23-year-old woman. It is surprising that nowhere in this review a clear differentiation of the various age groups of pediatric and adolescent (PAG) patients with adnexal tumors can be found. These groups are particularly important in adnexal masses ultrasound imaging when the frequency of various benign and malignant tumors varies substantially. In the presented reference of the study of Oltmann et al. [2010] the investigators have found that female patients aged 1 to 8 years have the greatest incidence of ovarian malignancy. The Author’s statement on reported frequency of adnexal malignancies mentioned as 10% up to 30% in paediatric and adolescent patients seems extremely exaggerated. For instance, Lam et al [ 2018] and Heo et al. [2014] have reported that ovarian malignancy could be found in 3–8% of children and adolescents diagnosed with an adnexal mass. Again, the Author’s statement “Up to 65% of ovarian masses in this age population are neoplastic” is hard to understand.

Response 2: Thank you for pointing this out. We agree with these comments.

However, the present study is a narrative review of the imaging findings of ovarian masses in the pediatric population, aiming to identify imaging features which, when combined with clinical characteristics and laboratory findings, can significantly aid in lesion characterization.

We used the term «pediatric population» in the revised version of the manuscript, based on the following definition: Pediatric malignancies” are defined as those malignancies commonly associated with the pediatric population that may arise during childhood, adolescence or young adulthood. [Long-Term Follow-Up Guidelines for Survivors of Childhood, Adolescent, and Young Adult Cancers. Version 6.0 (October 2023)].

Based on the above, we have included two cases of ovarian masses in young women.

It was not within the scope of this review to perform a separate analysis of imaging data for ovarian masses across different age groups within the pediatric population (children, adolescents, and young women).

We revised the comment on the incidence of ovarian malignancies in pediatric population as follows :

Most ovarian tumors in children and adolescents are benign, but malignancies are observed in 3-8 % of children and adolescents with an adnexal mass.

 We deleted the following phrase:

Up to 65% of ovarian masses in this age population are neoplastic

Comment 3: In the Introduction section the Authors have also cited an annual rate of 2,6 cases per 100.000 patients. However, it is not clear what did they really mean by citing this figure. This piece of information did not refer to a specific time, population and/or country. Moreover, a close look at the cited supporting references numbered 1-9 reveals that none of them has been based on large epidemiologic studies. The Authors should consider discussing examples of the relevant population-based retrospective cohort studies that would analyse the age groups of girls ≤18 years with ovarian/adnexal masses only. Moreover, to make the review meaningful, the discrimination of such masses should rather be addressed in separate groups of patients, for example in purely pediatric population, like in neonates, and in prepubertal and postpubertal patients. The prevalence of ovarian masses in these groups could then be meaningfully compared to the frequency of ovarian lesions found in the  “young” women category. However, no such information is presented in the reviewed manuscript. 

Response 3:  

However, the present study is a narrative review of the imaging findings of ovarian masses in the pediatric population, aiming to identify imaging features which, when combined with clinical characteristics and laboratory findings, can significantly aid in lesion characterization.  It is not within the scope of this review to perform a separate analysis of imaging data for ovarian masses across different age groups within the pediatric population (children, adolescents, and young women).

The following comment: “Ovarian tumors are rare in pediatric population, with an annual incidence of 2.6 cases per 100.000 girls” was based on Refs 1,2.

Comment 4: Second, I do not understand why the Authors have chosen to present only a median tumor size for various groups of adnexal masses. The median size without range assessment of various types of tumors is virtually useless in a single case of ovarian lesion diagnostics. The Authors should consider to add information on the tumor size range in mm, or like in the case of the medians, also add upper and lower quartiles for each group of ovarian lesions. For specific types of masses, like e.g. serous and mucinous cystadenomas it would also be more valuable to present  both the median diameter and range, i.e the smallest and the largest tumor size that have been reported for each group. 

Response 4: Thank you for pointing this out. However, in the relevant published literature, information about the size of different ovarian tumors in pediatric population is typically reported as the mean diameter. We chose to include mean size as an additional feature that may aid in differential diagnosis—for example, in the case of teratomas, where the mean diameter of immature teratomas is approximately 16 cm, compared to 6.5 cm for mature teratomas. Similarly, mean size can help identify large tumors exceeding 10 cm, such as JGCTs.

Comment 5: Third, the reader would certainly be interested on how the US morphology of adnexal/ovarian masses were described. In particular it would be interesting to present not only the  propoprtion of  unilateral and bilateral  lesions but also how many tumors in each group were unilocular, multilocular, unilocular-solid, multilocular-solid as well as purely solid.  Next, tumor vascularity as seen on color/power Doppler sonography should be presented. In such cases, the description should mention the existing and well-established criteria, like for instance the IOTA group COLOR score of ovarian lesion assessment. 

Response 5: Thank you for pointing this out. Regarding the morphology of different histopathologic subgroups of ovarian tumors in the pediatric population, we aimed to provide a diagnostic approach based on the typical imaging appearances of ovarian masses that may aid in differential diagnosis. A detailed description of the specific morphologic characteristics in each subgroup is beyond the scope of the present study, as it is not a meta-analysis but a narrative review. Since the imaging findings of different ovarian tumors in children, adolescents, and young women can be non-specific and overlapping, we aimed to present the most common imaging features, which, when combined with clinical and laboratory findings, can assist in lesion characterization.

Tumor vascularity characteristics on CDUS were added when they were missing  

For the use of IOTA scoring in pediatric populations see response 7.

Comment 6: Fourth, it is not clear from the presented data if the Authors were aware of the data on the proportion of ovarian tumors in each particular group that before surgery were correctly classified by the original ultrasound examiner. If such information is currently not available for a paticular ovarian tumor type, at least the most typical features for each separate group in pediatric and adolescent population should be presented. By using multiple citations on the ultrasound discrimination of ovarian tumors, the Authors admit that in this respect their “young women” category does not differ from the data cited for the “adult” female population. 

Response 6: Thank you for pointing this out. The imaging findings of the ovarian masses in the pediatric population are presented both in the Text and in Tables 1-3. However, it is not within the scope of this review to perform a separate analysis of imaging data for ovarian masses across different age groups within the pediatric population (children, adolescents, and young women).

Comment 7: Fifth, the Authors should also consider to comment on the currently used methods of ovarian masses discrimination, not only on presented “pattern recognition”, i.e. expert assessment of tumor morphology, but also on other existing scoring systems like: RMI, ROMA or IOTA group models. If, to date, such information has not been available for particular age groups and specific ovarian masses , it should be clearly stated. The same problem is related to other maging studies like CT or MRI in adolescent patients.

Response 7: Thank you for pointing this out. We agree with this comment. Comments on the scoring systems were added (Lines : 102-142 and Lines : 954-959)

Comment 8: It is not clear why the Authors firmly suggest that the radiologists should use the ORADS-US system for ovarian masses in pediatric population when they also admit that this system has not been validated in this age group. They state (lines 95-96), quote “Pediatric radiologists should apply the Ovarian-Adnexal Reporting and Data US System (O-RADS US) when assessing an ovarian lesion…”.

Response 8: Thank you for pointing this out. We agree with this comment. Revisions were made accordingly (Lines : 132-142).

Comment 9: Although the recent important study is mentioned as a citation, there is no detailed discussion with the data presented by Cooper et al.[2024]. These Authors have attempted not only to describe normative imaging findings and adnexal masses features in pediatric and adolescent populations, but also they have compared the performance of expert opinion alone, as well as using retrospective application of the International Ovarian Tumor Analysis (IOTA) simple rules (SRs) and benign descriptors (BDs) in those found to have an adnexal mass. More importantly, their analysis was performed on three different age groups: neonatal (aged < 1 year), premenarchal (aged ≥ 1 year) and postmenarchal up to 21 years old.

Response 9: Thank you for pointing this out. We agree with this comment. Revisions were made accordingly (Lines : 119-131).

Comment 10: Sixth, in the Introduction (lines 54-55) the Authors state that, quote: “Common non-neoplastic ovarian lesions in children and young females include functional cysts, endometriomas…” It may be controversial, but typically endometriomas are related to an abnormal growth of the ovarian tissues and because of that these lesions should be classified as a form of neoplastic growth within the ovary. Moreover, these masses, like breast or endometrial cancer, depend on estrogenic hormonal stimulation, and, although rarely, may mimic the ovarian malignancy or over the years may transform into malignancy. Taking into account these considerations, endometriomas should rather be described as benign neoplasms and not as “non-neoplastic ovarian lesions”. According to the WHO 2020 classification endometrial tumors, both benign or malignant, belong to the category “ovarian epithelial neoplasms” and certainly do not belong to “non-neoplastic lesions” (former group 9). Therefore the “endometrioma” subchapter should be placed in 3.3.1 category (starting in line 779) that describes “benign epithelial ovarian neoplasms”.

Response 10: Thank you for pointing this out. We agree with this comment. Revisions were made accordingly (Lines : 912-935).

Comment 11: Seventh, the table that starts on page 28 and extends to page 36 is apparently too long, contains too many data and because of these reasons it is very difficult to read. A summary for each tumor type or tumor groups in separate tables would be more comprehensive.

Response 11: Thank you for pointing this out. We agree with this comment. Separate tables (Tables 1-3) for the main histologic categories of ovarian tumors in the pediatric population were added. Revisions were made accordingly (Lines : 119-131).

Comment 12: Finally, the Author’s “Conclusions” section is a brief summary of existing textbook-type opinions and does not contain any new knowledge on the presented topic. Very similar conclusions have been already presented in older and newer systematic reviews referring to imaging and diagnostic issues in female pediatric and adolescent patients with ovarian masses. The Authors should rather consider different approach and conclude which groups of ovarian masses deserve multimodality imaging, including US, MRI or CT. It is apparent that not all, but only few adolescent patients would need such a complex and costly approach. Certainly, multicenter and larger studies are required for the prospective validation of imaging features and diagnostic models in PAGs with ovarian masses. Such new and prospectively validated strategies could aid an appropriate patient’s referral and enable to select the tailored patients care including fertility-sparing approach.

Response 12: Thank you for pointing this out. We agree with this comment. Revisions were made accordingly in the Conclusion section. A part on Future Directions was also added.

Reviewer 2 Report

Comments and Suggestions for Authors

Thank you for the opportunity to review the manuscript. It is very well-written, comprehensive and worth publishing review. However, some amendment are required.

Please find below my comments.

  1. The title is not attractive. The authors may think of something more informative.
  2. The manuscript is lacing information of scoring systems used for ovarian mass evaluation and based on imaging methods.
  3. Suggested review doi: 10.1080/01443615.2020 that could be helpful and provide an ideas of white kind of scoring systems are missing in the current manuscript. 
  4. Include a section on ROMA, ROCA, IOTA and other scoring systems utilization and value.

Author Response

Thank you for the opportunity to review the manuscript. It is very well-written, comprehensive and worth publishing review. However, some amendment are required.

Please find below my comments.

  1. The title is not attractive. The authors may think of something more informative.

Response 1: Thank you for pointing this out. We agree with this comment. The Title was changed as follows : Imaging Evaluation of Ovarian Masses in Pediatric Population: A Comprehensive Overview

2. The manuscript is lacing information of scoring systems used for ovarian mass evaluation and based on imaging methods.

Response 2: Thank you for pointing this out. We agree with this comment. Revisions were made accordingly (Lines 102-142).

3. Suggested review doi: 10.1080/01443615.2020 that could be helpful and provide an ideas of white kind of scoring systems are missing in the current manuscript. 

Response 3: Thank you for pointing this out. We agree with this comment. The review was added in the Refs list.

4. Include a section on ROMA, ROCA, IOTA and other scoring systems utilization and value.

Response 4: Thank you for pointing this out. We agree with this comment. Revisions were made accordingly (Lines 102-142).

Reviewer 3 Report

Comments and Suggestions for Authors

Dear Authors,
 You performed a nice and  comprehensive review on main imaging findings in ovarian tumoral pathology in pediatric population . Although you did not bring any  news and no real scientific apport , I consider that your work has some value and can be published.
I suggest you to include tables summarizing the main immaging findings on each ovarian mass.
Also, do you have patient consent and ethical approval for the immages in the manuscript? 

Author Response

Dear Authors,
You performed a nice and  comprehensive review on main imaging findings in ovarian tumoral pathology in pediatric population. Although you did not bring any news and no real scientific apport, I consider that your work has some value and can be published.

Comment 1: I suggest you to include tables summarizing the main imaging findings on each ovarian mass.

Response 1: Thank you for pointing this out. We agree with this comment. Separate tables (Tables 1-3) for the main histologic categories of ovarian tumors in the pediatric population were added.

Comment 2: Also, do you have patient consent and ethical approval for the images in the manuscript? 

Response 2: Thank you for pointing this out. We have patient consent and ethical approval for the images.

Round 2

Reviewer 1 Report

Comments and Suggestions for Authors

The Authors have addressed most of the critical problems that I have pointed out and made adequate corrections to their revised manuscript. Some minor issues that should in my opinion be commented on and/or corrected are listed below.

The Authors claim that, quote: “they have used the term «pediatric population» in the revised version of the manuscript, based on the following definition: Pediatric malignancies” are defined as those malignancies commonly associated with the pediatric population that may arise during childhood, adolescence or young adulthood. [Long-Term Follow-Up Guidelines for Survivors of Childhood, Adolescent, and Young Adult Cancers. Version 6.0 (October 2023)].”

Although this may be taken as an explanation, still it is very strange to compare young adult females with preadolescent/adolescent girls and/or female newborn population.

The Authors state quote: “Based on the above, we have included two cases of ovarian masses in young women.”. Although this statement could be taken as an explanation, but to the best of my knowledge, the definition cited above has not been widely accepted in the scientific research. I think that these two cases should be specially commented on and/or removed from the manuscript. The description should be clearly separated, as in my opinion they have nothing in common with adolescent girls and clearly belong to the adult (although “young”) women age group.

I cannot agree with the Author’s reply and the statement, quote: “It was not within the scope of this review to perform a separate analysis of imaging data for ovarian masses across different age groups within the pediatric population (children, adolescents, and young women).” These groups differ significantly in many aspects like e.g. hormonal status (prepubertal and postbubertal, pre- and post-menarcheal) and also in terms of various ovarian malignancies prevalence that is related with patient’s age. Combining such different age groups leads to inevitable comparisons of various data categories like e.g. apples and oranges- both are fruits, but they differ significantly in many aspects. It is plausible, that some or multiple imaging features of different malignant ovarian masses differ in various pediatric and adolescent age groups, but we do not know this yet. The presented and revised narrative review does not add any new data to resolve and/or discuss this problem.

The Authors have not commented sufficiently the issue of presented mean ovarian tumor sizes. In particular to support their statement that ovarian tumor size has been found to be an independent predictor of malignancy they quote a reference #2. Although these guidelines summarized by Behr et al in 2023 may be used by pediatricians, in my opinion by no means cannot guide imaging studies results and clinical assessment of the risk of malignancy. The Authors reply, quote: ” We chose to include mean size as an additional feature that may aid in differential diagnosis—for example, in the case of teratomas, where the mean diameter of immature teratomas is approximately 16 cm, compared to 6.5 cm for mature teratomas. Similarly, mean size can help identify large tumors exceeding 10 cm, such as JGCTs.“ still is not satisfactory. First, in a clinical setting of the adolescent female patient information on suspected ovarian tumor size by any means cannot be taken as the main category used to distinguish  lesion type, i.e. malignant vs. benign. This may be misleading as for example in cases of non-palpable borderline ovarian epithelial tumors. These lesions despite their small size (i.e. maximum size<5cm, or even <3cm) on imaging studies frequently present typical set of suspicious for malignancy features, like e.g. single or multiple papillary projections in their internal wall. Second, imaging features of very rare in all female age groups malignant teratomas do not resemble the most frequent benign variants of these ovarian lesions and because of that, their size remains a very controversial prognostic feature. Third, although tumor size is included in several clinically validated prognostic models like e.g. the extremely efficient polytomous ADNEX model developed in 2014 by the IOTA group, still small malignant tumors presenting with typically low CA125 serum levels are more likely to be inadequately classified on imaging studies. This in turn could lead to fatal diagnostic and therapeutic mistakes. Therefore, in my opinion, an information on mean various adnexal tumor sizes should be critically addressed in any narrative or systematic review on the presented topic.  Here, the Authors may consider to add in this paragraph, or in the Discussion section a statement like e.g. “Although ovarian tumor size may play a role in initial risk of malignancy assessment in pediatric population, it should be emphasized that at least some small masses, especially presenting solid-cystic and/or solid morphology as seen on imaging studies and regardless of serum tumor markers expression can also turn out to be malignant.”

Author Response

The Authors have addressed most of the critical problems that I have pointed out and made adequate corrections to their revised manuscript. Some minor issues that should in my opinion be commented on and/or corrected are listed below.

 Comments 1: [The Authors claim that, quote: “they have used the term «pediatric population» in the revised version of the manuscript, based on the following definition: Pediatric malignancies” are defined as those malignancies commonly associated with the pediatric population that may arise during childhood, adolescence or young adulthood. [Long-Term Follow-Up Guidelines for Survivors of Childhood, Adolescent, and Young Adult Cancers. Version 6.0 (October 2023)].”

Although this may be taken as an explanation, still it is very strange to compare young adult females with preadolescent/adolescent girls and/or female newborn population.

The Authors state quote: “Based on the above, we have included two cases of ovarian masses in young women.”. Although this statement could be taken as an explanation, but to the best of my knowledge, the definition cited above has not been widely accepted in the scientific research. I think that these two cases should be specially commented on and/or removed from the manuscript. The description should be clearly separated, as in my opinion they have nothing in common with adolescent girls and clearly belong to the adult (although “young”) women age group]

Response 1: Thank you for pointing this out. We agree with this comment.

Therefore, we have revised our manuscript, including children (ages 0 to 14 years) and adolescents (ages 15-19 years), based on the definition of the ACS facts & figures 2025 (https://www.cancer.org/content/dam/cancer-org/research/cancer-facts-and-statistics/annual-cancer-facts-and-figures/2025/2025-cancer-facts-and-figures-acs.pdf). (Lines 49,50).

We have also deleted the figures showing cases in young females.

Comments 2: [I cannot agree with the Author’s reply and the statement, quote: “It was not within the scope of this review to perform a separate analysis of imaging data for ovarian masses across different age groups within the pediatric population (children, adolescents, and young women).” These groups differ significantly in many aspects like e.g. hormonal status (prepubertal and postbubertal, pre- and post-menarcheal) and also in terms of various ovarian malignancies prevalence that is related with patient’s age. Combining such different age groups leads to inevitable comparisons of various data categories like e.g. apples and oranges- both are fruits, but they differ significantly in many aspects. It is plausible, that some or multiple imaging features of different malignant ovarian masses differ in various pediatric and adolescent age groups, but we do not know this yet. The presented and revised narrative review does not add any new data to resolve and/or discuss this problem].

Response 2: Agree. We have revised our manuscript, including children (ages 0 to 14 years) and adolescents (ages 15-19 years), based on the definition of the ACS facts & figures 2025 (https://www.cancer.org/content/dam/cancer-org/research/cancer-facts-and-statistics/annual-cancer-facts-and-figures/2025/2025-cancer-facts-and-figures-acs.pdf).

Comments 3: [The Authors have not commented sufficiently the issue of presented mean ovarian tumor sizes. In particular to support their statement that ovarian tumor size has been found to be an independent predictor of malignancy they quote a reference #2. Although these guidelines summarized by Behr et al in 2023 may be used by pediatricians, in my opinion by no means cannot guide imaging studies results and clinical assessment of the risk of malignancy. The Authors reply, quote: ” We chose to include mean size as an additional feature that may aid in differential diagnosis—for example, in the case of teratomas, where the mean diameter of immature teratomas is approximately 16 cm, compared to 6.5 cm for mature teratomas. Similarly, mean size can help identify large tumors exceeding 10 cm, such as JGCTs.“ still is not satisfactory. First, in a clinical setting of the adolescent female patient information on suspected ovarian tumor size by any means cannot be taken as the main category used to distinguish  lesion type, i.e. malignant vs. benign. This may be misleading as for example in cases of non-palpable borderline ovarian epithelial tumors. These lesions despite their small size (i.e. maximum size<5cm, or even <3cm) on imaging studies frequently present typical set of suspicious for malignancy features, like e.g. single or multiple papillary projections in their internal wall. Second, imaging features of very rare in all female age groups malignant teratomas do not resemble the most frequent benign variants of these ovarian lesions and because of that, their size remains a very controversial prognostic feature. Third, although tumor size is included in several clinically validated prognostic models like e.g. the extremely efficient polytomous ADNEX model developed in 2014 by the IOTA group, still small malignant tumors presenting with typically low CA125 serum levels are more likely to be inadequately classified on imaging studies. This in turn could lead to fatal diagnostic and therapeutic mistakes. Therefore, in my opinion, an information on mean various adnexal tumor sizes should be critically addressed in any narrative or systematic review on the presented topic.  Here, the Authors may consider to add in this paragraph, or in the Discussion section a statement like e.g. “Although ovarian tumor size may play a role in initial risk of malignancy assessment in pediatric population, it should be emphasized that at least some small masses, especially presenting solid-cystic and/or solid morphology as seen on imaging studies and regardless of serum tumor markers expression can also turn out to be malignant.”]

Response 3: Thank you for pointing this out. We agree with this comment. The statement was included in the revised version of the manuscript (Lines 101-104].

Reviewer 3 Report

Comments and Suggestions for Authors

Dear Authors,

Thank you for your answers. The manuscript is improved. I consider it suitable for publication.

Best regards.

Author Response

Comments: [Dear Authors,

Thank you for your answers. The manuscript is improved. I consider it suitable for publication.

Best regards]

Response: Thank you very much for taking the time to review this manuscript.

.